# Synthesis and Antimicrobial Activity of Phosphonopeptide Derivatives Incorporating Single and Dual Inhibitors

**DOI:** 10.3390/molecules25071557

**Published:** 2020-03-28

**Authors:** Keng Tiong Ng, John D. Perry, Emma C. L. Marrs, Sylvain Orenga, Rosaleen J. Anderson, Mark Gray

**Affiliations:** 1Sunderland Pharmacy School, University of Sunderland, Sunderland SR1 3SD, UK; keng_tiong.ng@kcl.ac.uk; 2Microbiology Department, Freeman Hospital, High Heaton, Newcastle upon Tyne NE7 7DN, UK; john.perry@nuth.nhs.uk (J.D.P.); e.marrs@nhs.net (E.C.L.M.); 3R&D Microbiologie, bioMérieux, 3 Route de Port Michaud, 38390 La Balme-les-Grottes, France; sylvain.orenga@biomerieux.com

**Keywords:** bacterial detection, suicide substrates, fosfalin, β-chloroalanine

## Abstract

In diagnostic microbiology, culture media are widely used for detection of pathogenic bacteria. Such media employ various ingredients to optimize detection of specific pathogens such as chromogenic enzyme substrates and selective inhibitors to reduce the presence of commensal bacteria. Despite this, it is rarely possible to inhibit the growth of all commensal bacteria, and thus pathogens can be overgrown and remain undetected. One approach to attempt to remedy this is the use of “suicide substrates” that can target specific bacterial enzymes and selectively inhibit unwanted bacterial species. With the purpose of identifying novel selective inhibitors, six novel phosphonopeptide derivatives based on d/l-fosfalin and β-chloro-l-alanine were synthesized and tested on 19 different strains of clinically relevant bacteria. Several compounds show potential as useful selective agents that could be exploited in the recovery of several bacterial pathogens including *Salmonella*, *Pseudomonas aeruginosa*, and *Listeria*.

## 1. Introduction

Antimicrobial resistance (AMR) is rapidly becoming a global crisis. According to the Centers for Disease Control and Prevention (CDC), around 2.8 million people are infected with antibiotic-resistant bacteria and approximately 35,000 deaths occur each year in the USA due to these infections [1,2]. Moreover, the O’Neill report estimates that there will be 10 million deaths globally each year due to antibiotic-resistant bacteria by 2050 (O’Neill’s AMR report, 2016). This impending crisis has reinforced the need for new, rapid, economical and user-friendly bacterial detection and identification methods. These tools will prove crucial in the years ahead in order to facilitate antibiotic stewardship, particularly in the selection of the most effective antibiotic to control specific pathogenic bacteria.

Chromogenic and fluorogenic enzymatic substrates are established diagnostic tools used in clinical microbiology for pathogen identification [3,4]. Used alone however, these diagnostic tools face two major problems: i) The presence of abundant commensal bacteria within clinical samples can camouflage the detection of low amounts of pathogenic bacteria, and ii) production of similar enzymes by different strains or species that act upon the same enzymatic substrate can lead to false results. Due to this, incorporation of specific and selective growth inhibitors such as l-alanyl-l-fosfalin **1** (Figure 1) into the growth medium is of clinical interest [5], as such substances may inhibit the growth of interfering microbes while allowing a pathogen of interest to grow unhindered. Once inside bacterial cells, cleavage of the peptide bond of l-alanyl-l-fosfalin occurs, releasing the active substance l-fosfalin **2-L** (Figure 1). This in turn binds to alanine racemase (AlaR), a bacterial enzyme that catalyzes the isomerization of l-alanine to D-alanine **3-D** (Figure 1) utilizing a pyridoxal 5′-phosphate cofactor [6]. As D-alanine is a crucial building block for bacterial cell wall formation [7], bacterial growth is hindered if the enzyme is inhibited. The expression of l-alanine aminopeptidase for peptide bond cleavage is found more readily in Gram-negative than Gram-positive bacteria [8]. In addition, some bacteria either do not take in the pseudo-dipeptide or it is removed rapidly by efflux pumps [9]. The combination of these factors provides a basis for selective inhibition of bacterial cell growth in culture.

Although fosfalin is a known inhibitor, its use as a single pseudo-amino acid residue is limited due to the multiply ionised forms of fosfalin found at physiologically relevant pH values. This factor reduces its ability to penetrate the cell membrane by passive diffusion. However, dipeptide inhibitors, such as l-alanyl-l-fosfalin, can bypass the cell membrane by hijacking di/tripeptide transporter systems found within the cell membrane [10].

Low minimum inhibitory concentrations (MICs) were observed upon replacing the l-alanine with different hydrophobic l-amino acids, especially l-methionine and l-norvaline at the *N*-terminus of phosphonodipeptide derivatives containing fosfalin [11,12]. Moreover, selective inhibition against *Escherichia coli*, *Klebsiella* species and *Enterococcus faecalis* has previously been found by replacing the l-alanine at the *N*-terminus of phosphonotripeptides with sarcosine [13].

Elsewhere, it has been shown that the alanine analogue β-chloroalanine **4** (Figure 1) is known to display irreversible inhibition towards alanine racemase [14]. In studies performed and reviewed by Atherton et al. [11,12,13] and Cheung et al. [15,16] it has been shown that β-chloroalanine exhibits good antibacterial activity alone as well as upon coupling with other peptides or amino acids.

Although many studies have been performed in the past based on fosfalin or β-chloroalanine peptide derivatives, to the best of our knowledge, no studies on the antimicrobial effects of molecules incorporating both of these inhibitors into a single peptide-like derivative have been reported. Herein, a range of phosphonotripeptides based either on a single inhibitor, d/l-fosfalin, or containing dual inhibitors, d/l-fosfalin and β-chloro-l-alanine, coupled to a variety of different *N*-terminal units in the form of sarcosine, l-norvaline or l-methionine were synthesized. The antibacterial activity of the resulting phosphonotripeptides was evaluated against 19 different strains of clinically relevant bacteria. Although l-fosfalin is most commonly studied version of this particular inhibitory unit, it is costly to produce. This factor may limit its use in the developing nations which lie on the front-line of the battle against emergent drug resistant bacteria. With this in mind, the racemic version of this unit, d/l-fosfalin was utilized in our studies, due to its relatively facile, and thus economical, production. 

## 2. Results and Discussion

### 2.1. Synthesis of Inhibitors: d/l-Fosfalin and β-Chloro-l-alanine Derivatives

Several synthetic methodologies producing fosfalin are known, including production of its enantiomerically pure forms [17,18,19,20]. Here however, synthesis of racemic fosfalin, i.e., d/l-fosfalin, was utilized due to its more economical synthesis. In our previous unpublished investigations, we have seen that whenever the enantiomerically pure form of fosfalin has been utilized within a peptide sequence the activity is always twice that of the analogous compound containing the racemic version of fosfalin. This is in keeping with the notion that one isomer of fosfalin is active and the other is completely inactive. As the intended use in the present work is to produce economically viable diagnostic tools rather than potent antimicrobial agents, we elected to utilize the racemic version of this warhead within this study. To that end, a good yield of d/l-fosfalin was obtained in one-pot, using N-phenylthiourea **5**, acetaldehyde **6** and triphenyl phosphite **7** under acidic conditions (Scheme 1). From there, two different protecting groups, *N*-trifluoroacetyl and an *O*-phosphonodiethyl ester group were introduced orthogonally at the *N*-terminus and *O*-terminus of racemic fosfalin respectively. Upon removal of *N*-trifluoroacetyl protecting group **8** by sodium borohydride, d/l-fosfalin diethyl ester **9** was isolated.

Producing the enantiomerically pure β-chloro-l-alanine derivatives, Boc-β-chloro-l-alanine **13** and β-chloro-l-alanine benzyl ester hydrochloride salt **14** was a more economical proposition than the fosfalin unit. Thus, these materials were prepared in their enantiomerically pure forms from an inexpensive and readily available chiral precursor, Boc-l-serine **10**. To produce the required molecules, the precursor was subjected to base-catalyzed esterification, followed by chlorination and then chemoselective deprotection (Scheme 2). The corresponding intermediates, Boc-l-Ser-OBzl **11** and Boc-β-Cl-l-Ala-OBzl **12** were isolated in good yield.

### 2.2. Synthesis of Phosphonotripeptide Derivatives

d/l-fosfalin diethyl ester **9** was coupled to three dipeptides producing a series of tripeptides with a central alanine residue and a variable *N*-terminal residue (Scheme 3). For this position we chose the amino acids that Atherton et al. [11,12,13] had indicated produced interesting effects in terms of bacterial cell permeation and MIC enhancement, namely sarcosine (Sar), methionine (Met), and norvaline (Nva). In a second series, l-alanine was replaced by β-chloro-l-alanine at the central position thus producing a series that contained two inhibitors within each tripeptide strand. In total, this resulted in six different phosphonotripeptide derivatives, l-Nva-l-Ala-d/l-Fos **21a**, Sar-l-Ala-d/l-Fos **21b**, l-Met-l-Ala-d/l-Fos **21c**, l-Nva-β-chloro-l-Ala-d/l-Fos **25a**, Sar-β-chloro-l-Ala-d/l-Fos **25b**, l-Met-β-chloro-l-Ala-d/l-Fos **25c**. The ratio of diastereoisomers and enantiomers can be found within the Appendix A.

A different coupling strategy was utilized where the reactive amino acid methionine, **15c**, was employed (Scheme 3, Pathway B). This amino acid displays a sensitive thioether side chain, which is prone to oxidation and cyclisation reactions [21]. Therefore, in order minimize these unwanted side reactions, methionine was introduced at the last stage of the synthesis of tripeptides (**20d** and **20e**).

### 2.3. Microbiological Evaluation

The inhibitory action of these compounds requires efficient uptake of these molecules across the bacterial cell membrane, hydrolysis of specific peptide bonds by aminopeptidases to release the inhibitor(s), and binding of inhibitor(s) to AlaR. As different amino acids are utilized at the *N*-terminus of the tripeptide derivatives, differences in uptake may result, and different intracellular aminopeptidases may be required for the liberation of d/l-fosfalin and β-chloro-l-alanine (where incorporated), in order to act on AlaR. The rationale for varying the *N*-terminal amino acid was that as different species and strains of bacteria produce the transporters and enzymes involved with these processes to differing extents, differences in antibacterial activity were anticipated. The antimicrobial activity of phosphonotripeptide derivatives was evaluated as their minimum inhibitory concentration (MIC), Table 1.

Broadly speaking, all six phosphonopeptide derivatives produced similar inhibitory profiles to one another against most bacterial species and strains. However, the extent of activity for the collection of inhibitors varied markedly depending on the bacterium in question in a manner that does not appear to be specifically tied to the Gram classification of the organism. For instance, consistently high activity is seen for all inhibitors for two of the Gram-negative species *S. marcescens* and *Y. enterocolitica*, with all inhibitors displaying MIC values of 2 mg/L or less. In contrast, no activity against six of the remaining Gram-negative organisms tested was observed at the highest inhibitor concentration tested, which was 8 mg/L. Turning attention to the Gram-positive organisms low or no activity was displayed at the maximum concentrations tested for *L. monocytogenes*, while very high activity (0.063 mg/L or less) was seen against *E. faecalis* for all inhibitors tested except for **21b** and **25b**, which still retained activity, albeit at the 4 mg/L level.

In general, compounds **21b** and **25b** appear to be the least effective in terms of overall inhibitory profile. In both instances, the *N*-terminal amino acid is sarcosine, thus the presence of this amino acid in this position of these tripeptide derivatives appears to be detrimental to activity. However, the current data set does not definitively explain whether these poor activity profiles derive from poor uptake, or from slow liberation of the active units once inside the bacterial cell. In many instances these two poorest performing inhibitors have identical MIC values, and when differences do exist they are displayed only to the nearest increment that our tests allow. For example, for *S. marcecsens* the substance with a simple alanine **21b** produces the greater inhibition (1 mg/L) over its β-chloro analogue **25b** (2 mg/L). In contrast, for *E. coli* NCTC 12,241 the reverse behavior is seen with the β-chloro substance **25b** exhibiting MIC = 4 mg/L, whereas in this case the simple alanine containing tripeptide **21b** produces a weaker MIC of 8 mg/L. Essentially, this indicates that these two poorly performing inhibitors display very similar levels of activity over a wide range of organisms.

With regards to the remaining four inhibitors, there is a subtle yet clear difference in activity for the central β-Cl-Ala series vs. the series of compounds with a standard alanine residue at the central position. Specifically, **25a** and **25c** are the most consistently active pair of inhibitors, and contain β-Cl-Ala at the center position of each tripeptide. Thus, this indicates that the presence of the β-chloro unit in the center of the tripeptide is indeed beneficial to overall activity. While not definitive, this would indicate that the β-chloro alanine unit is liberated, and becomes active against the bacteria, acting in a synergistic manner with the fosfalin unit, which is common to all six inhibitors.

l-Alanine aminopeptidase is thought to be found more readily in Gram-negative bacteria [8]. However, in our data some inhibitory effects extended to certain Gram positive bacteria, namely *E. faecalis*, *E. faecium*, *S. epidermidis*, *S. aureus,* and methicillin resistant *S. aureus* (MRSA). This suggests the likely presence of l-alanine aminopeptidase in these selected Gram-positive bacteria. This finding was consistent with results recently reported by Cellier et al. [23], who demonstrated a chromogenic response upon hydrolysis of l-alanyl aminopeptidase substrates by *E. faecalis* and *E. faecium*. The presence of this enzymatic activity in *Staphylococcus* spp., *Streptococcus* spp. and *E. faecalis* has also been reported by Hoosain and Lastovica [24].

As intimated earlier, the eventual purpose of these compounds was to be incorporated into selective culture media for the clinical diagnosis of bacterial infections. Thus, we seek to identify substrates that have a wide range of inhibitory activities but leave specific pathogens of interest free to grow. Looking at our current data set, this suggests a potential application for two of our β-chloro-l-alanyl phosphonotripeptides (**25a** and **25c**) in the selective inhibition of specific Gram-negative bacteria. Specifically, *E. cloacae*, *E. coli*, *K. pneumoniae*, and *S. marcescens* are commonly found in cystic fibrosis samples [25,26,27]. Our results indicate that these can be selectively inhibited, and thus prevent overgrowth of *B. cepacia* and *P. aeruginosa* which often cause severe, and sometimes fatal, lung infections in these patients. A second potential application of our work would be for detection of *Salmonella* spp. in the clinic. This is due to the relatively weak inhibitory levels (>8 mg/L) displayed by our compounds against these organisms in contrast to the more effective inhibition against the growth of *E. coli* (<4 mg/L). This is clinically useful as *E. coli* often overgrow *Salmonella* spp. when isolated from stool samples [5]. A further potential application of such compounds is the inhibition of *Enterococcus* spp. facilitating the isolation of *Listeria monocytogenes* and pathogenic streptococci e.g., *Streptococcus pyogenes*.

In this paper we present preliminary microbiology data to identify novel compounds that may prove useful as novel selective agents in diagnostic culture media. A major limitation of our data is that only 1 or 2 examples of any particular species have been tested and susceptibility (or resistance) may not be uniformly demonstrated across all strains of a particular species. The utility of these compounds can only be proven in subsequent studies by testing large numbers of strains from each relevant species and large numbers of clinical samples to see if the isolation of target pathogens may be enhanced. However, the fact that such utility as selective agents has been demonstrated for other peptide mimetics based on fosfalin provides encouragement that such agents may ultimately prove useful [5,28].

## 3. Materials and Methods

All commercially available reagents and solvents were acquired from Sigma-Aldrich, Fluka, Alfa-aesar, Fluorochem, Fischer Scientific and Bachem and used without further purification unless otherwise stated. All deuterated solvents were purchased from Goss Scientific and Apollo Scientific. Solvents were dried using the solvent purification system, PureSolv 400-5-MD, which was purchased from Innovative Technology (currently known as Inert) (Amesbury, MA, USA). Hydrogenation was performed using series 4560 mini reactors, acquired from Parr Instruments (Moline, Il, USA). Column chromatography was performed using Silica 60A (35–70 or 70–200 micron) Davisil^®^ chromatography grade (Merck, Loughborough, UK). Thin layer chromatography was performed on Merck TLC Silica Gel 60 F254 (Loughborough, UK) or Fluka Silca (Dorset, UK) on TLC Alu foils. Melting points were recorded using a Reichart-Kofler (Staffordshire, UK) hot stage microscope apparatus and are uncorrected. Infrared spectra were recorded in the range 4000–550 cm^−1^ using a Perkin Elmer Spectrum BX FT-IR spectrophotometer (Waltham, MA, USA). NMR spectra were acquired using a Bruker Ultrashield 300 spectrometer (Billerica, MA, USA) at 300 MHz for ^1^H spectra, 75 MHz for ^13^C spectra, 121 MHz for ^31^P-^1^H decoupled spectra and 282 MHz for ^19^F-^1^H decoupled spectra. NMR data was analyzed by MestReNova (Mestrelab Research, Santiago de Compostela, Spain). Low-resolution mass spectra were obtained from a Bruker Esquire 3000plus analyzer (Billerica, MA, USA) using an electrospray source in either positive or negative ion mode. High-resolution mass spectra were acquired from Thermo Scientific LTQ Orbitrap XL (Waltham, MA, USA) using an electrospray source in either positive or negative ion mode. Elemental analyses were performed using an Exeter Analytical CE-440 Elemental Analyzer (Coventry, UK). The LC column, ACE Excel 5 Super C18 (150 × 4.6 mm) and Eclipse Plus C18 (50 × 2.1 mm) were acquired from HICHROM (Reading, UK) and Agilent (Waldbronn, Germany), respectively. Elemental and purity by LC-MS were performed on final compounds or compounds which analytical data are not available in literature.

### 3.1. Synthetic Procedures

#### 3.1.1. General IBCF/NMM Peptide Coupling Method

In 100 mL one neck round-bottomed flask A, *Cα*-protected amine salt (1.0 equiv) or phosphonate ester protected amine salt (when l-Met was used) was dissolved in dry DCM or dry THF (25–50 mL) and cooled to −5 °C in an ice/brine bath. To this was added *N,N*-diisopropylethylamine, DIPEA (1.5 equiv) or *N*-methyl morpholine, NMM (1.0 equiv) (when β-Cl-l-Ala was used). In 250 mL one neck round-bottomed flask B, *Nα*-protected amino acid (1.0 equiv) was dissolved in dry THF (25–50 mL) and to this was added *N*-methyl morpholine, NMM (1.5 equiv or 1.0 equiv when β-Cl-l-Ala moiety was involved) and isobutyl chloroformate (1.0 equiv) while stirring at −5 °C. The Flask B solution was stirred for 2–3 h at −5 °C, after which time the neutralized *Cα*-protected or phosphonate ester protected amine solution in Flask A was then added. The resulting mixture was stirred at −5 °C for 10mins and then room temperature overnight (16–18 h). Work-up procedure: the mixture was filtered and filtrate was concentrated in vacuo to afford a crude product, which was dissolved in DCM and washed with 10 %w/v citric acid (2 × 30 mL), followed by 10 %w/v potassium carbonate (30 mL) and finally water (30 mL). The organic layer was dried over magnesium sulfate, filtered, concentrated in vacuo, and purified by column chromatography using an appropriate solvent system to afford the product of interest.

#### 3.1.2. Removal of Benzyl Ester Protecting Group

*Cα*-Benzyl ester amino acid or peptide was dissolved in methanol and to this was added 5%–10% palladium on carbon (10 mol %). The mixture was stirred at 3.5 bar pressure of hydrogen at room temperature overnight (16–18 h). The catalyst was filtered using Celite and the filtrate was concentrated in vacuo to afford the product of interest.

#### 3.1.3. Removal of *N*-tert-Butoxycarbonyl Protecting Group

*N*-*^t^*Boc amino acid or peptide was dissolved in excess of a solution of dry 2M HCl in diethyl ether. The resulting solution was stirred at room temperature for 2 days to afford a crude solid, which was collected by filtration and washed with excess dry diethyl ether to afford the product of interest.

#### 3.1.4. Removal of tert-Butoxycarbonyl and Diethyl Phosphonate Ester Protecting Groups

Phosphonopeptide containing *N-tert*-butoxycarbonyl and *O,O*-diethyl phosphonate ester was dissolved in 35 %wt HBr in acetic acid (10 equiv) and stirred at room temperature overnight (16–18 h). Dry diethyl ether was added to afford a precipitation. The resulting mixture was stored in the freezer overnight (16–18 h) to result in more precipitation. The solvent was decanted and remaining crude solid was triturated (×3) with dry diethyl ether. The resulting solid was dissolved in dry methanol and propylene oxide was added in excess to afford a crude precipitation, which was recrystallized from the appropriate solvent system to afford the product of interest. 


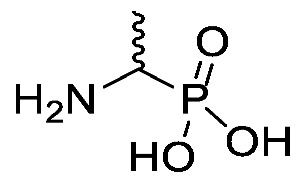


*1-Aminoethylphosphonic Acid or d/l-fosfalin* (**2-DL**). To suspension of *N*-phenylthiourea (100.0 mmol, 15.2 g) in glacial acetic acid (50 mL), acetaldehyde (130.0 mmol, 7.40 mL) was added dropwise, followed by triphenyl phosphite (100.0 mmol, 27.0 mL). The mixture was stirred at room temperature for 5 mins, then refluxed at 80 °C for 1 h until a clear solution was obtained. A mixture of glacial acetic acid (5 mL) and hydrochloric acid (37%, 50 mL) was added and the reaction was refluxed overnight. The solution was cooled to room temperature and concentrated in vacuo to afford a brown slurry. Absolute ethanol (150 mL) was added while stirring and the resulting off-white solid was collected by filtration and dried in a desiccator containing phosphorus(V) oxide. The crude solid was recrystallized from hot water/ethanol to afford **2-DL** as white crystals, as a mixture of enantiomers (12.2 g, 98 mmol, 98%); m.p. 271–274 °C (sublim); *ῡ*_max_/cm^−1^ 2910 (br OH), 1532 (NH bend), 1143 (P=O), 1035 (P-O-C), 930 (P-OH); ^1^H NMR (300 MHz, D_2_O) δ_H_ 1.40 (3H, dd, ^3^*J*_H-P_ = 14.7 Hz, ^3^*J*_H-H_ = 7.2 Hz, C**H_3_**), 3.33 (1H, m, C**H**); ^13^C NMR (75 MHz, D_2_O) δ_C_ 13.5 (d, ^2^*J*_C-P_ = 2.6 Hz, **C**H_3_), 44.7 (d, ^1^*J*_C-P_ = 144.2 Hz, **C**H); ^31^P-^1^H_decoup_ NMR (121 MHz, D_2_O) δ_P_ 14.2; *m/z* (ESI) calcd for (C_2_H_9_NO_3_P)^+^, MH^+^: 126.0, found 126.1; CHN (Found: C, 19.45; H, 6.48; N, 11.18. C_2_H_8_NO_3_P requires C, 19.21; H, 6.45; N, 11.20%). (The ^1^H- and ^13^C-NMR spectra may be found within the Appendix A).


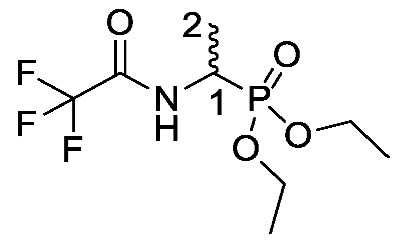


*Diethyl (1-(2,2,2-trifluoroacetamido)ethyl)phosphonate or trifluoroacetyl-d/l-Fos diethyl ester* (**8**). 1-Aminoethylphosphonic acid (**2-DL**) (51.7 mmol, 6.5 g) was added to a mixture of trifluoroacetic acid (65.3 mmol, 5 mL) and trifluoroacetic anhydride (177.4 mmol, 25 mL). The solution was stirred and refluxed at 60 °C for 1 h, then cooled to room temperature and triethyl orthoformate (901.8 mmol, 150 mL) was added dropwise. The solution was refluxed at 110 °C for 2 h, then cooled to room temperature. The solution was concentrated in vacuo to afford a brown solid, which was re-dissolved in DCM and purified by column chromatography using a gradient elution (DCM (100) to DCM/MeOH (95:5)) to give **8** as an off-white solid, a mixture of enantiomers (11.4 g, 41.0 mmol, 80%); m.p. 101–103 °C (sublim) (lit. m.p. 101–102 °C) [29]; *ῡ*_max_/cm^−1^ 3202 (NH), 1715 (C=O), 1565 (NH bend), 1210 (P=O), 1011 (P-O-C), 968 (P-O-C); ^1^H NMR (300 MHz, CDCl_3_) δ_H_ 1.24 (3H, t, ^3^*J*_H-H_ = 7.2 Hz, OCH_2_C**H_3_**), 1.27 (3H, t, ^3^*J*_H-H_ = 7.2 Hz, OCH_2_C**H_3_**), 1.38 (3H, dd, ^3^*J*_H-P_ = 16.5 Hz, ^3^*J*_H-H_ = 7.2 Hz, C**H_3_**-2), 4.06 (4H, m, 2 × OC**H_2_**CH_3_), 4.39 (1H, m, C**H**-1), 8.00 (1H, d, ^3^*J*_H-H_ = 6.0 Hz, N**H**); ^13^C NMR (75 MHz, CDCl_3_) δ_C_ 14.8 (**C**H_3_-2), 16.2 (d, ^3^*J*_C-P_ = 2.3 Hz, OCH_2_**C**H_3_), 16.3 (d, ^3^*J*_C-P_ = 2.3 Hz, OCH_2_**C**H_3_), 41.8 (d, ^1^*J*_C-P_ = 159.1 Hz, **C**H-1), 62.8 (d, ^2^*J*_C-P_ = 7.0 Hz, O**C**H_2_CH_3_), 63.2 (d, ^2^*J*_C-P_ = 7.1 Hz, O**C**H_2_CH_3_), 115.9 (q, ^1^*J*_C-F_ = 285.8 Hz, **C**F_3_), 156.9 (q, ^2^*J*_C-F_ = 5.8 Hz, **C**=O); ^31^P-^1^H_decoup_ NMR (121 MHz, CDCl_3_) δ_P_ 23.0; ^19^F-^1^H_decoup_ NMR (282 MHz, CDCl_3_) δ_P_ −75.5; *m/z* (ESI) calcd for (C_8_H_16_F_3_NO_4_P)^+^, MH^+^: 278.1, found 278.1.


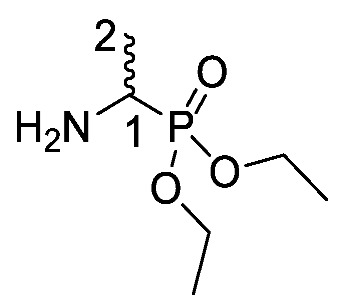


*Diethyl 1-aminoethylphosphonate or d/l-Fos diethyl ester* (**9**). Diethyl (1-(2,2,2-trifluoroacetamido)ethyl)phosphonate (**8**) (20.0 mmol, 5.6 g) was dissolved in ethanol (200 ml) and excess sodium borohydride (200.0 mmol, 7.7 g) was added slowly with stirring. The resulting mixture was stirred at room temperature for 1 h, then heated at reflux for 4 h. The mixture was cooled to room temperature and the solvent was removed *in vacuo* to afford a white solid, which was dissolved in saturated NaHCO_3_ (96 g/L) (60 mL) with the addition of 10% aqueous K_2_CO_3_ (20 mL). The product was extracted into DCM (6 × 30 mL) and dried over MgSO_4_. The filtrate was concentrated *in vacuo* to afford a pale yellow liquid and purified by column chromatography using a gradient elution (DCM (100) to DCM/MeOH (90:10)) to afford **9** as a yellow liquid, a mixture of enantiomers (3.5 g, 19.3 mmol, 97%); *ῡ*_max_/cm^−1^ 3431 (NH), 1215 ( P=O), 1020 (P-O-C), 967 (P-O-C); ^1^H NMR (300 MHz, CDCl_3_) δ_H_ 1.26 (6H, t, ^3^*J*_H-H_ = 7.2 Hz, 2 × OCH_2_C**H_3_**), 1.34 (3H, dd, ^3^*J*_H-P_ = 17.7 Hz, ^3^*J*_H-H_ = 7.2 Hz, C**H_3_**-2), 1.68 (2H, br, N**H_2_**), 3.02–3.12 (1H, m, C**H**-1), 4.06–4.17 (4H, m, 2 × OC**H_2_**CH_3_); ^13^C NMR (75 MHz, CDCl_3_) δ_C_ 16.4 (OCH_2_**C**H_3_), 16.5 (OCH_2_**C**H_3_), 17.2 (**C**H_3_-2), 44.2 (d, ^1^*J*_C-P_ = 148.5 Hz, **C**H-1), 62.1 (d, ^2^*J*_C-P_ = 7.5 Hz, O**C**H_2_CH_3_), 62.1 (d, ^2^*J*_C-P_ = 7.5 Hz, O**C**H_2_CH_3_); ^31^P-^1^H_decoup_ NMR (121 MHz, CDCl_3_) δ_P_ 29.6; HRMS (NSI) calcd for (C_6_H_17_NO_3_P)^+^, MH^+^: 204.0760, found 204.0762. LCMS purity >95% (C-18 reversed phase, MeOH–H_2_O).


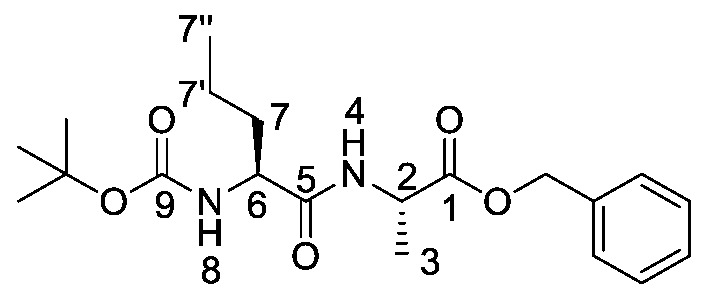


*(S)-Benzyl 2-((S)-2-((tert-butoxycarbonyl)amino)pentanamido) propanoate or Boc-l-Nva-l-Ala-OBzl* (**16a**). General peptide coupling method was followed, using Boc-l-Nva-OH (**15a**) (10.0 mmol, 2.17 g) in dry THF and l-alanine benzyl ester *p*-tosylic acid (10.0 mmol, 3.52 g) in dry DCM. The yellow crude liquid was purified by column chromatography (40–60 petrol/ethyl acetate (7:3)) to give **16a** as an off-white solid (2.40 g, 6.3 mmol, 63%); m.p. 60–63 °C; *ῡ*_max_/cm^−1^ 3332 (NH), 1743 (C=O), 1655 (br C=O), 1527 (NH bend), 1245 (C-O), 1162 (C-O); ^1^H NMR (300 MHz, CDCl_3_) δ_H_ 0.83 (3H, t, ^3^*J*_H-H_ = 9.0 Hz, C**H_3_**-7″), 1.25–1.31 (2H, m, C**H_2_**-7′), 1.34 (3H, d, ^3^*J*_H-H_ = 6.0 Hz, C**H_3_**-3), 1.36 (9H, s, C(C**H_3_**)_3_), 1.42–1.54 (1H, m, C**H_a/b_**-7), 1.64–1.73 (1H, m, C**H_a/b_**-7), 4.02 (1H, m, C**H**-6), 4.54 (1H, pentet, ^3^*J*_H-H_ = 6.0 Hz, C**H**-2), 4.96 (1H, d, ^3^*J*_H-H_ = 9.0 Hz, N**H**-8), 5.07 (1H, d, ^2^*J*_H-H_ = 12.0 Hz, OC**H_a/b_**Ar), 5.12 (1H, d, ^2^*J*_H-H_ = 12.0 Hz, O**CH_a/b_**Ar), 6.56 (1H, d, ^3^*J*_H-H_ = 6.0 Hz, N**H**-4), 7.27 (5H, m, 5 × C**H_Ar_**); ^13^C NMR (75 MHz, CDCl_3_) δ_C_ 12.7 (**C**H_3_-7″), 17.3 (**C**H_3_-3), 17.8 (**C**H_2_-7′), 27.3 (C(**C**H_3_)_3_), 33.7 (**C**H_2_-7), 47.1 (**C**H-2), 53.4 (**C**H-6), 66.1(O**C**H_2_Ar), 79.0 (**C**(CH_3_)_3_), 127.1-127.6 (**C**H_Ar_), 134.3 (**C**H_Ar_ quat.), 154.6 (**C**=O-9), 170.8 (**C**=O-5), 171.5 (**C**=O-1); HRMS (NSI) calcd for (C_20_H_31_N_2_O_5_)^+^, MH^+^: 379.2227, found 379.2222; CHN (Found: C, 63.75; H, 8.37; N, 7.86. C_20_H_30_N_2_O_5_ requires C, 63.47; H, 7.99; N, 7.40%).


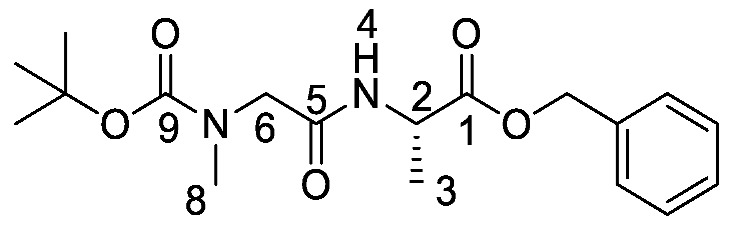


*(S)-Benzyl 2-(2-((tert-butoxycarbonyl)(methyl)amino)acetamido) propanoate or Boc-Sar-l-Ala-OBzl* (**16b**). General peptide coupling method was followed, using Boc-Sar-OH (**15b**) (15.0 mmol, 2.84 g) in dry THF and l-alanine benzyl ester *p*-tosylic acid (15.0 mmol, 5.27 g) in dry DCM. The yellow crude liquid was purified by column chromatography (40–60petrol/ethyl acetate (1:1)) to give **16b** as a colourless liquid (3.93 g, 11 mmol, 75%); *ῡ*_max_/cm^−1^ 3311 (NH), 1742 (C=O), 1670 (C=O), 1666 (C=O), 1536 (NH bend), 1242 (C-O), 1145 (C-O); ^1^H NMR (300 MHz, CDCl_3_) δ_H_ 1.35 (3H, t, ^3^*J*_H-H_ = 6.0 Hz, C**H_3_**-3), 1.39 (9H, s, C(C**H_3_**)_3_), 2.85 (3H, s, C**H_3_**-8), 3.72 (1H, d, ^2^*J*_H-H_ = 15.0 Hz, C**H_a/b_**-6), 3.88 (1H, d, ^2^*J*_H-H_ = 15.0 Hz, C**H_a/b_**-6), 4.58 (1H, pentet, ^3^*J*_H-H_ = 6.0 Hz, C**H**-2), 5.08 (1H, d, ^2^*J*_H-H_ = 12.0 Hz, OC**H_a/b_**Ar), 5.13 (1H, d, ^2^*J*_H-H_ = 12.0 Hz, O**CH_a/b_**Ar), 6.51 (1H, br, N**H**-4), 7.25-7.29 (5H, m, 5 × C**H_Ar_**); ^13^C NMR (75 MHz, CDCl_3_) δ_C_ 17.5 (**C**H_3_-3), 27.3 (C(**C**H_3_)_3_), 34.7 (**C**H_3_-8), 47.0 (**C**H-2), 52.1 (**C**H_2_-6), 66.2 (O**C**H_2_Ar), 79.8 (**C**(CH_3_)_3_), 127.1–127.6 (**C**H_Ar_), 134.3 (**C**H_Ar_ quat.), 155.0 (**C**=O-9), 167.9 (**C**=O-5), 171.5 (**C**=O-1); HRMS (NSI) calcd for (C_18_H_27_N_2_O_5_)^+^, MH^+^: 351.1914, found 351.1916. LCMS purity >95% (C-18 reversed phase, MeOH–H_2_O).


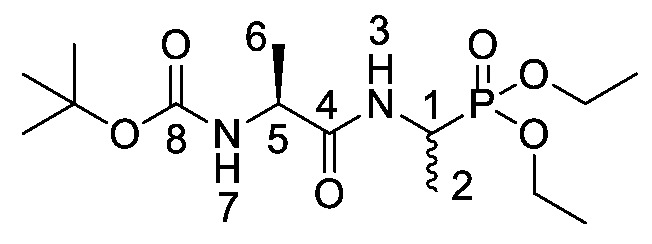


*Tert-butyl ((2S)-1-((-1-(diethoxyphosphoryl)ethyl)amino)-1-oxopropan-2-yl)carbamate or Boc-l-Ala-d/l-Fos diethyl ester* (**19d**). General peptide coupling method was followed, using Boc-l-Ala-OH (**15d**) (10.0 mmol, 1.90 g) in dry THF and diethyl 1-aminoethylphosphonate (**9**) (10.0 mmol, 1.84 g) in dry THF. The pale yellow crude syrup was purified by column chromatography, using 100% DCM and increasing to 95:5 DCM/methanol, to afford **19d** as an off-white solid composed of 2 diastereoisomers, Boc-l-Ala-l-Fos diethyl ester and Boc-l-Ala-D-Fos diethyl ester (2.49 g, 7.1 mmol, 71%); m.p. 102–105 °C; *ῡ*_max_/cm^−1^ 3280 (NH), 1710 (C=O), 1652 (C=O), 1556 (NH bend), 1229 (P=O), 1173 (C-O), 1013 (P-O-C), 973 (P-O-C); ^1^H NMR (300 MHz, CDCl_3_) δ_H_ 1.23–1.43 (12H, m, C**H_3_**-2, C**H_3_**-6, 2 × OCH_2_C**H_3_**), 1.44 (9H, s, C(C**H_3_**)_3_), 4.06–4.23 (5H, m, 2 × OC**H_2_**CH_3_, C**H**-5), 4.40–4.52 (1H, m, C**H**-1), 5.12 (0.5H, d, ^3^*J*_H-H_ = 1.5 Hz, N**H**-7), 5.14 (0.5H, d, ^3^*J*_H-H_ = 1.5 Hz, N**H**-7), 6.72 (0.5H, d, ^3^*J*_H-H_ = 2.3 Hz, N**H**-3), 6.74 (0.5H, d, ^3^*J*_H-H_ = 2.3 Hz, N**H**-3); ^13^C NMR (75 MHz, CDCl_3_) δ_C_ 15.6 (**C**H_3_-2), 16.3 (d, ^3^*J*_C-P_ = 3.0 Hz, OCH_2_**C**H_3_), 16.4 (d, ^3^*J*_C-P_ = 2.3 Hz, OCH_2_**C**H_3_), 16.5 (d, ^3^*J*_C-P_ = 3.0 Hz, OCH_2_**C**H_3_), 16.6 (d, ^3^*J*_C-P_ = 2.3 Hz, OCH_2_**C**H_3_), 18.4 (**C**H_3_-6), 28.3 (C(**C**H_3_)_3_), 40.8 (d, ^1^*J*_C-P_ = 156.8 Hz, **C**H-1), 41.0 (d, ^1^*J*_C-P_ = 156.8 Hz, **C**H-1), 50.0 (**C**H-5), 62.4 (d, ^2^*J*_C-P_ = 6.8 Hz, O**C**H_2_CH_3_), 62.5 (d, ^2^*J*_C-P_ = 6.8 Hz, O**C**H_2_CH_3_), 62.6 (d, ^2^*J*_C-P_ = 6.8 Hz, O**C**H_2_CH_3_), 62.8 (d, ^2^*J*_C-P_ = 6.8 Hz, O**C**H_2_CH_3_), 80.0 (**C**(CH_3_)_3_), 155.2 (**C**=O-8), 172.1 (**C**=O-4); ^31^P-^1^H_decoup_ NMR (121 MHz, CDCl_3_) δ_P_ 25.2; HRMS (NSI) calcd for (C_17_H_35_N_3_O_7_P)^+^, MH^+^: 424.2207, found 424.2200; CHN (Found: C, 48.22; H, 8.58; N, 7.87. C_14_H_29_N_2_O_6_P requires C, 47.92; H, 8.30; N, 7.95%).


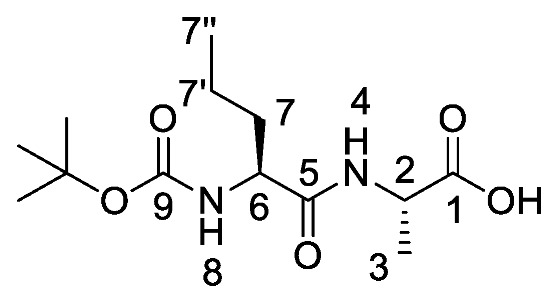


*(S)-2-((S)-2-((Tert-butoxycarbonyl)amino)pentanamido) propanoic acid or Boc-l-Nva-l-Ala-OH* (**17a**). Deprotection of benzyl ester was followed, using (*S*)-benzyl 2-((*S*)-2-((*tert*-butoxycarbonyl)amino)pentanamido)propanoate (**16a**) (6.0 mmol, 2.27 g) to afford **17a** as a white solid (1.66 g, 5.7 mmol, 96.0%); m.p. 55–58 °C (decomp.); *ῡ*_max_/cm^−1^ 3500–3000 (br, OH), 3300 (NH), 1688 (br C=O), 1655 (C=O), 1522 (NH bend), 1245 (C-O), 1164 (C-O); ^1^H NMR (300 MHz, CDCl_3_) δ_H_ 0.85 (3H, t, ^3^*J*_H-H_ = 9.0 Hz, C**H_3_**-7″), 1.27–1.31 (5H, m, C**H_3_**-3, C**H_2_**-7′), 1.39 (9H, s, C(C**H_3_**)_3_), 1.48–1.53 (1H, m, C**H_a/b_**-7), 1.67–1.71 (1H, m, C**H_a/b_**-7), 4.10 (1H, m, C**H**-6), 4.50 (1H, m, C**H**-2), 5.27 (1H, m, N**H**-8), 6.93 (1H, m, N**H**-4), 8.87 (1H, br, O**H**); ^13^C NMR (75 MHz, CDCl_3_) δ_C_ 13.7 (**C**H_3_-7″), 18.0 (**C**H_3_-3), 18.8 (**C**H_2_-7′), 28.3 (C(**C**H_3_)_3_), 34.5 (**C**H_2_-7), 48.1 (**C**H-2), 54.3 (**C**H-6), 80.4 (**C**(CH_3_)_3_), 156.0 (**C**=O-9), 172.5 (**C**=O-5), 175.5 (**C**=O-1); HRMS (NSI) calcd for (C_13_H_25_N_2_O_5_)^+^, MH^+^: 289.1758, found 289.1758; CHN (Found: C, 54.18; H, 8.78; N, 9.62. C_13_H_24_N_2_O_5_ requires C, 54.15; H, 8.39; N, 9.72%).


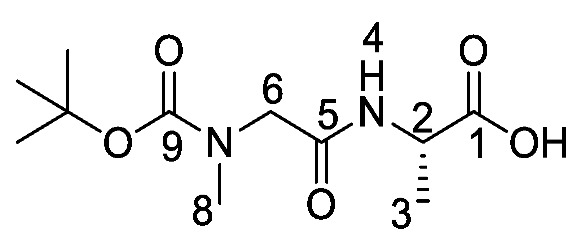


*(S)-Benzyl 2-(2-((tert-butoxycarbonyl)(methyl)amino)acetamido) propanoic acid or Boc-Sar-l-Ala-OH* (**17b**). Deprotection of benzyl ester was followed, using (*S*)-benzyl 2-(2-((*tert*-butoxycarbonyl)(methyl)amino)acetamido) propanoate (**16b**) (10.0 mmol, 3.51 g) to afford **17b** as a colorless syrup (2.50 g, 9.6 mmol, 96%); *ῡ*_max_/cm^−1^ 3301 (NH), 2961 (broad OH), 1736 (C=O), 1664 (br C=O), 1542 (NH bend), 1241 (C-O), 1147 (C-O); ^1^H NMR (300 MHz, CDCl_3_) δ_H_ 1.36 (3H, t, ^3^*J*_H-H_ = 6.0 Hz, C**H_3_**-3), 1.39 (9H, s, C(C**H_3_**)_3_), 2.89 (3H, s, C**H_3_**-8), 3.72 (1H, d, ^2^*J*_H-H_ = 18.0 Hz, C**H_a/b_**-6), 3.98 (1H, d, ^2^*J*_H-H_ = 18.0 Hz, C**H_a/b_**-6), 4.57 (1H, m, C**H**-2), 6.96 (1H, m, N**H**-4), 7.26 (1H, br, O**H**); ^13^C NMR (75 MHz, CDCl_3_) δ_C_ 17.2 (**C**H_3_-3), 27.3 (C(**C**H_3_)_3_), 46.8 (**C**H-2), 49.6 (**C**H_3_-8), 52.1 (**C**H_2_-6), 80.6 (**C**(CH_3_)_3_), 155.5 (**C**=O-9), 168.3 (**C**=O-5), 174.1 (**C**=O-1); HRMS (NSI) calcd for (C_11_H_19_N_2_O_5_)^-^, MH^−^: 259.1299, found 259.1295. LCMS purity >95% (C-18 reversed phase, MeOH-H_2_O).


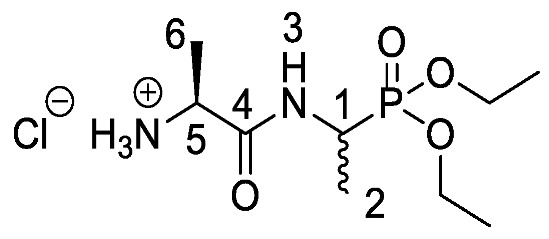


*(2S)-1-((1-(Diethoxyphosphoryl)ethyl)amino)-1-oxopropan-2-aminium chloride or l-Ala-d/l-Fos diethyl ester hydrochloride* (**20d**). Deprotection of tert-butoxycarbonyl was followed, using tert-butyl ((2S)-1-((-1-(diethoxyphosphoryl)ethyl)amino)-1-oxopropan-2-yl)carbamate (**19d**) (6.0 mmol, 2.13 g). The off-white hygroscopic crude solid was washed with petrol to afford **20d** as a pale green solid composed of 2 diastereoisomers, l-Ala-l-Fos diethyl ester hydrochloride and l-Ala-D-Fos diethyl ester hydrochloride (1.46 g, 5.1 mmol, 84%); m.p. 60–63 °C; ῡ_max_/cm^−1^ 2986 (NH^+^), 1673 (C=O), 1555 (NH bend), 1212(P=O), 1017 (P-O-C), 970 (P-O-C); ^1^H NMR (300 MHz, CD_3_OD) δ_H_ 1.29-1.44 (9H, m, 2 × OCH_2_C**H_3_**, C**H_3_**-2), 1.51 (3H, d, ^3^*J*_H-H_ = 6.0 Hz, C**H_3_**-6), 3.90-3.98 (1H, m, C**H**-5), 4.08-4.22 (4H, m, 2 × OC**H_2_**CH_3_), 4.28-4.47 (1H, m, C**H**-1); ^13^C NMR (75 MHz, CD_3_OD) δ_C_ 13.7 (**C**H_3_-2), 14.0 (**C**H_3_-2), 15.4 (2 × OCH_2_**C**H_3_), 16.3 (**C**H_3_-6), 41.1 (d, ^1^*J*_C-P_ = 158.3 Hz, **C**H-1), 41.4 (d, ^1^*J*_C-P_ = 158.3 Hz, **C**H-1), 48.8 (**C**H-5), 48.9 (**C**H-5), 62.7–63.0 (2 × O**C**H_2_CH_3_), 169.0 (**C**=O-4); ^31^P-^1^H_decoup_ NMR (121 MHz, CDCl_3_) δ_P_ 29.0, 29.1; HRMS (NSI) calcd for (C_9_H_22_N_2_O_4_P)^+^, M^+^: 253.1312, found 253.1316. LCMS purity >95% (C-18 reversed phase, MeOH-H_2_O).


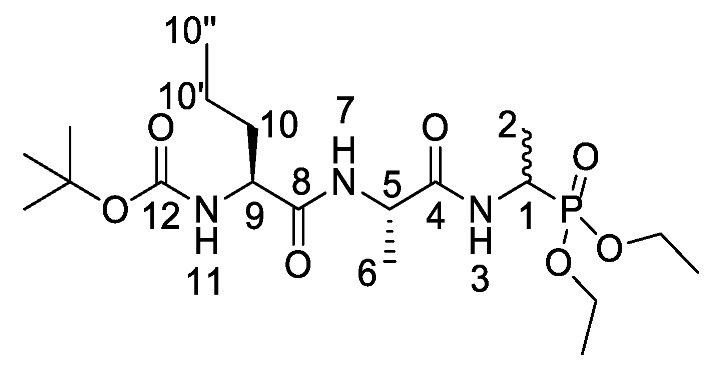


*Tert-butyl ((2S)-1-(((2S)-1-((1-(diethoxyphosphoryl)ethyl) amino)-1-oxopropan-2-yl)amino)-1-oxopentan-2-yl)carbamate or Boc-l-Nva-l-Ala-d/l-Fos diethyl ester* (**18a**). General peptide coupling method was followed, using (S)-2-((S)-2-((tert-butoxycarbonyl)amino)pentanamido)propanoic acid (**17a**) (5.0 mmol, 1.45 g) in dry THF and diethyl 1-aminoethylphosphonate (**9**) (4.8 mmol, 0.87 g) in dry THF. The white crude solid was purified by column chromatography using 100% DCM, increasing to 90:10 DCM/methanol, to afford **18a** as a white solid composed of 2 diastereoisomers, Boc-l-Nva-l-Ala-l-Fos diethyl ester and Boc-l-Nva-l-Ala-D-Fos diethyl ester (1.70 g, 3.8 mmol, 78%); m.p. 165–168 °C; ῡ_max_/cm^−1^ 3267 (NH), 1708 (C=O), 1638 (br C=O), 1537 (NH bend), 1227 (P=O), 1165 (C-O), 1019 (P-O-C), 966 (P-O-C); ^1^H NMR (300 MHz, CDCl_3_) δ_H_ 0.85 (3H, t, ^3^*J*_H-H_ = 9.0 Hz, C**H_3_**-10″), 1.18–1.34 (14H, m, 2 × OCH_2_C**H_3_**, C**H_3_**-2, C**H_3_**-6, C**H_2_**-10′), 1.37 (9H, s, C(C**H_3_**)_3_), 1.47–1.54 (1H, m, C**H_a/b_**-10), 1.65–1.73 (1H, m, C**H_a/b_**-10), 3.98–4.12 (5H, m, 2 × OC**H_2_**CH_3_, C**H**-9), 4.33–4.44 (1H, m, C**H**-1), 4.48–4.54 (1H, m, C**H**-5), 5.19 (0.5H, d, ^3^*J*_H-H_ = 6.0 Hz, N**H**-11), 5.23 (0.5H, d, ^3^*J*_H-H_ = 6.0 Hz, N**H**-11), 6.78 (0.5H, d, ^3^*J*_H-H_ = 6.0 Hz, N**H**-7), 6.87 (0.5H, d, ^3^*J*_H-H_ = 6.0 Hz, N**H**-7), 7.15 (0.5H, d, ^3^*J*_H-H_ = 9.0 Hz, N**H**-3), 7.23 (0.5H, d, ^3^*J*_H-H_ = 9.0 Hz, N**H**-3); ^13^C NMR (75 MHz, CDCl_3_) δ_C_ 12.7 (**C**H_3_-10″), 14.4 (**C**H_3_-2), 14.5 (**C**H_3_-2), 15.3 (OCH_2_**C**H_3_), 15.4 (OCH_2_**C**H_3_), 15.5 (OCH_2_**C**H_3_), 15.6 (OCH_2_**C**H_3_), 17.6 (**C**H_3_-6), 17.7 (**C**H_3_-6), 17.8 (**C**H_2_-10′), 17.9 (**C**H_2_-10′), 27.3 (C(**C**H_3_)_3_), 33.8 (**C**H_2_-10), 33.9 (**C**H_2_-10), 39.9 (d, ^1^*J*_P-C_ = 157.5 Hz, **C**H-1), 40.0 (d, ^1^*J*_P-C_ = 156.8 Hz, **C**H-1), 47.7 (**C**H-5), 47.9 (**C**H-5), 53.5 (**C**H-9), 53.5 (**C**H-9), 61.5 (d, ^2^*J*_C-P_ = 7.5 Hz, O**C**H_2_CH_3_), 61.6 (d, ^2^*J*_C-P_ = 7.5 Hz, O**C**H_2_CH_3_), 61.7 (d, ^2^*J*_C-P_ = 7.5 Hz, O**C**H_2_CH_3_), 61.9 (d, ^2^*J*_C-P_ = 7.5 Hz, O**C**H_2_CH_3_), 78.9 (**C**(CH_3_)_3_), 154.7 (C=O-12), 170.6 (**C**=O-4 or **C**=O-8), 170.7 (**C**=O-4 or **C**=O-8), 171.0 (**C**=O-4 or **C**=O-8), 171.1 (**C**=O-4 or **C**=O-8); ^31^P-^1^H_decoup_ NMR (121 MHz, CDCl_3_) δ_P_ 25.0, 25.1; HRMS (NSI) calcd for (C_19_H_39_N_3_O_7_P)^+^, MH^+^: 452.2520, found 452.2518; CHN (Found: C, 50.74; H, 8.55; N, 9.51. C_19_H_38_N_3_O_7_P requires C, 50.54; H, 8.48; N, 9.31%). 


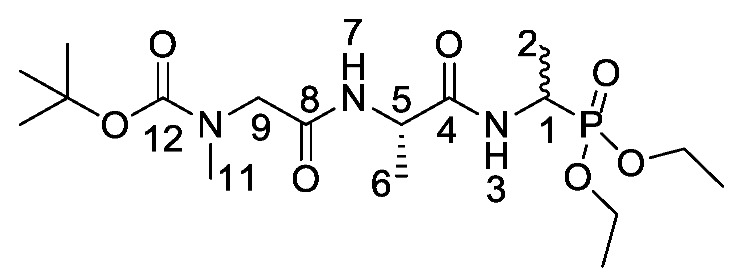


*Tert-butyl (2-(((2S)-1-((1-(diethoxyphosphoryl)ethyl)amino)-1-oxopropan-2-yl)amino)-2-oxoethyl)(methyl)carbamate or Boc-Sar-l-Ala-d/l-Fos* (**18b**). General peptide coupling method was followed, using (S)-benzyl 2-(2-((tert-butoxycarbonyl)(methyl)amino)acetamido)propanoic acid (**17b**) (6.0 mmol, 1.57 g) in dry THF and diethyl 1-aminoethylphosphonate (**9**) (6.0 mmol, 1.10 g) in dry THF. The yellow crude liquid was purified by column chromatography, using 100% DCM and increasing to 90:10 DCM/methanol, to afford **18b** as a colorless liquid composed of 2 diastereoisomers, Boc-Sar-l-Ala-l-Fos diethyl ester and Boc-Sar-l-Ala-D-Fos diethyl ester (1.60 g, 3.8 mmol, 63%); ῡ_max_/cm^−1^ 3270 (NH), 1700 (br C=O), 1655 (C=O), 1545 (NH bend), 1225 (P=O), 1149 (C-O), 1018 (P-O-C), 966 (P-O-C); ^1^H NMR (300 MHz, CDCl_3_) δ_H_ 1.19–1.34 (12H, m, C**H_3_**-2, C**H_3_**-6, 2 × OCH_2_C**H_3_**), 1.40 (9H, s, C(C**H_3_**)_3_), 2.87 (3H, s, C**H_3_**-11), 3.72 (0.5H, d, ^2^*J*_H-H_ = 15.0 Hz, C**H_a/b_**-9), 3.78 (0.5H, d, ^2^*J*_H-H_ = 15.0 Hz, C**H_a/b_**-9), 3.81 (0.5H, d, ^2^*J*_H-H_ = 15.0 Hz, C**H_a/b_**-9), 3.87 (0.5H, d, ^2^*J*_H-H_ = 15.0 Hz, C**H_a/b_**-9), 4.00–4.11 (4H, m, 2 × OC**H_2_**CH_3_), 4.35-4.43 (1H, m, C**H**-1), 4.47–4.52 (1H, m, C**H**-5), 6.67 (1H, d, ^3^*J*_H-H_ = 9.0 Hz, N**H**-7), 6.98 (0.5H, d, ^3^*J*_H-H_ = 9.0 Hz, N**H**-3), 7.15 (0.5H, d, ^3^*J*_H-H_ = 9.0 Hz, N**H**-3); ^13^C NMR (75 MHz, CDCl_3_) δ_C_ 15.5 (**C**H_3_-2), 15.5 (**C**H_3_-2), 16.3 (d, ^3^*J*_P-C_ = 3.0 Hz, OCH_2_**C**H_3_), 16.4 (d, ^3^*J*_P-C_ = 3.0 Hz, OCH_2_**C**H_3_), 18.7 (**C**H_3_-6), 28.3 (C(**C**H_3_)_3_), 35.8 (**C**H_3_-11), 41.0 (d, ^1^*J*_P-C_ = 157.5 Hz, **C**H-1), 48.5 (**C**H-5), 53.0 (**C**H_2_-9), 62.5 (d, ^2^*J*_P-C_ = 6.8 Hz, O**C**H_2_CH_3_), 62.6 (d, ^2^*J*_P-C_ = 6.8 Hz, O**C**H_2_CH_3_), 62.7 (d, ^2^*J*_P-C_ = 6.8 Hz, O**C**H_2_CH_3_), 62.9 (d, ^2^*J*_P-C_ = 6.8 Hz, O**C**H_2_CH_3_), 80.7 (**C**(CH_3_)_3_), 156.0 (C=O-12), 171.5 (**C**=O-4 or **C**=O-8), 171.6 (**C**=O-4 or **C**=O-8); ^31^P-^1^H_decoup_ NMR (121 MHz, CDCl_3_) δ_P_ 25.0, 25.1; HRMS (NSI) calcd for (C_17_H_35_N_3_O_7_P)^+^, MH^+^: 424.2207, found 424.2203. LCMS purity >95% (C-18 reversed phase, MeOH–H_2_O).


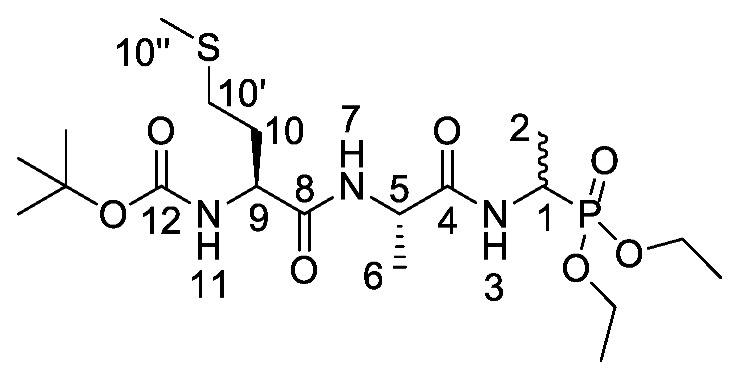


*Tert-butyl ((2S)-1-(((2S)-1-((1-(diethoxyphosphoryl)ethyl) amino)-1-oxopropan-2-yl)amino-4-(methylthio)-1-oxobutan-2-yl)carbamate or Boc-l-Met-l-Ala-d/l-Fos diethyl ester* (**18c**). General peptide coupling method was followed, using Boc-l-Ala-Met-OH (**15c**) (3.4 mmol, 0.88 g) in dry THF and (2S)-1-((1-(diethoxyphosphoryl)ethyl)amino)-1-oxopropan-2-aminium chloride (**20d**) (3.4 mmol, 0.97 g) in dry DCM. The yellow crude solid was purified by column chromatography (DCM/MeOH (95:5)) to give **18c** as an off-white solid composed of 2 diastereoisomers, Boc-l-Met-l-Ala-l-Fos diethyl ester and Boc-l-Met-l-Ala-D-Fos diethyl ester (0.53 g, 1.1 mmol, 32%); m.p. 172–176 °C; ῡ_max_/cm^−1^ 3272 (NH), 1708 (C=O), 1673 (C=O), 1637 (C=O), 1530 (NH bend), 1226 (P=O), 1165 (C-O), 1020 (P-O-C), 976 (P-O-C); ^1^H NMR (300 MHz, CDCl_3_) δ_H_ 1.16–1.36 (12H, m, C**H_3_**-2, C**H_3_**-6, 2 × OCH_2_C**H_3_**), 1.36 (9H, s, C(C**H_3_**)_3_), 1.82–2.01 (2H, m, C**H_2_**-10), 2.04 (3H, s, C**H_3_**-10″), 2.49 (2H, dd, ^3^*J*_H-H_ = 9.0 Hz, 3.0 Hz, C**H_2_**-10′), 4.00–4.12 (4H, m, 2 × OC**H_2_**CH_3_), 4.16–4.26 (1H, m, C**H**-9), 4.33–4.43 (1H, m, C**H**-1), 4.45–4.53 (1H, m, C**H**-5), 5.40 (0.5H, d, ^3^*J*_H-H_ = 9.0 Hz, N**H**-11), 5.44 (0.5H, d, ^3^*J*_H-H_ = 6.0 Hz, N**H**-11), 6.85 (0.5H, d, ^3^*J*_H-H_ = 6.0 Hz, N**H**-7), 6.92 (0.5H, d, ^3^*J*_H-H_ = 6.0 Hz, N**H**-7), 7.07 (0.5H, d, ^3^*J*_H-H_ = 9.0 Hz, N**H**-3), 7.16 (0.5H, d, ^3^*J*_H-H_ = 9.0 Hz, N**H**-3); ^13^C NMR (75 MHz, CDCl_3_) δ_C_ 14.2 (**C**H_3_-2), 14.3 (**C**H_3_-2), 14.5 (**C**H_3_-10″), 14.6 (**C**H_3_-10″), 15.4 (d, ^3^*J*_C-P_ = 3.0 Hz, OCH_2_**C**H_3_), 15.4 (d, ^3^*J*_C-P_ = 2.3 Hz, OCH_2_**C**H_3_), 15.5 (d, ^3^*J*_C-P_ = 3.0 Hz, OCH_2_**C**H_3_), 15.5 (d, ^3^*J*_C-P_ = 2.3 Hz, OCH_2_**C**H_3_), 17.7 (**C**H_3_-6), 27.3 (C(**C**H_3_)_3_), 29.2 (**C**H_2_-10′), 29.3 (**C**H_2_-10′), 30.8 (**C**H_2_-10), 30.9 (**C**H_2_-10), 39.9 (d, ^1^*J*_C-P_ = 156.8 Hz, **C**H-1), 40.0 (d, ^1^*J*_C-P_ = 156.8 Hz, **C**H-1), 47.9 (**C**H-5), 48.0 (**C**H-5), 52.6 (**C**H-9), 61.5 (d, ^2^*J*_C-P_ = 6.8 Hz, O**C**H_2_CH_3_), 61.6 (d, ^2^*J*_C-P_ = 6.8 Hz, O**C**H_2_CH_3_), 61.7 (d, ^2^*J*_C-P_ = 6.8 Hz, O**C**H_2_CH_3_), 61.9 (d, ^2^*J*_C-P_ = 6.8 Hz, O**C**H_2_CH_3_), 79.1 (**C**(CH_3_)_3_), 154.6 (**C**=O-12), 170.3 (**C**=O-4 or **C**=O-8), 170.4 (**C**=O-4 or **C**=O-8), 170.5 (**C**=O-4 or **C**=O-8), 170.6 (**C**=O-4 or **C**=O-8); ^31^P-^1^H_decoup_ NMR (121 MHz, CDCl_3_) δ_P_ 25.0, 25.1; HRMS (NSI) calcd for (C_19_H_39_N_3_O_7_PS)^+^, MH^+^: 484.2241, found 484.2228. LCMS purity >95% (C-18 reversed phase, MeOH–H_2_O).


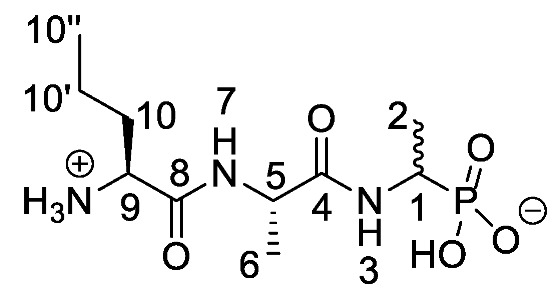


*(1-((S)-2-((S)-2-Aminopentanamido)propanamido)ethyl) phosphonic acid or l-Nva-l-Ala-d/l-Fos* (**21a**). The *tert*-butoxycarbonyl and diethyl ester protecting groups of *tert*-butyl ((2*S*)-1-(((2*S*)-1-((1-(diethoxyphosphoryl)ethyl)amino)-1-oxopropan-2-yl)amino)-1-oxopentan-2-yl)carbamate (**18a**) (1.6 mmol, 0.72 g) were removed. The pale green crude solid was recrystallised from hot water/acetone to give **21a** as a pale green solid composed of 2 diastereoisomers, l-Nva-l-Ala-l-Fos and l-Nva-l-Ala-D-Fos (0.22 g, 0.75 mmol, 47%); m.p. 207–210 °C (decomp.); *ῡ*_max_/cm^−1^ 3280 (NH^+^), 3500-2900 (br OH), 1643 (br C=O), 1552 (NH bend), 1149 (P=O), 1037 (P-O-C), 922 (P-OH); ^1^H NMR (300 MHz, D_2_O) δ_H_ 0.96 (3H, t, ^3^*J*_H-H_ = 7.1 Hz, CH_3_-10″), 1.27-1.32 (3H, d, ^3^*J*_H-H_ = 6.8 Hz, C**H_3_**-2), 1.40-1.42 (3H, m, C**H_3_**-6), 1.40–1.42 (2H, m, C**H_2_**-10′), 1.88–1.86 (2H, m, C**H_2_**-10), 4.00–4.02 (2H, m, C**H**-1, C**H**-9), 4.34–4.39 (1H, m, C**H**-5); ^13^C NMR (75 MHz, D_2_O) δ_C_ 13.4 (**C**H_3_-10″), 16.0 (**C**H_3_-2), 17.1 (**C**H_3_-6), 17.2 (**C**H_3_-6), 18.1 (**C**H_2_-10′), 18.2 (**C**H_2_-10′), 33.5 (**C**H_2_-10), 33.6 (**C**H_2_-10), 50.5 (**C**H-5), 50.8 (**C**H-5), 53.5 (**C**H-1, **C**H-9), 170.4 (**C**=O-8), 170.6 (**C**=O-8),174.7 (**C**=O-4); ^31^P-^1^H_decoup_ NMR (121 MHz, CDCl_3_) δ_P_ 18.5; HRMS (NSI) calcd for (C_10_H_23_N_3_O_5_P)^+^, MH^+^: 296.1370, found 296.1373. LCMS purity >95% (C-18 reversed phase, MeOH-H_2_O). (The LCMS chromatogram and conditions may be found within the Appendix A).


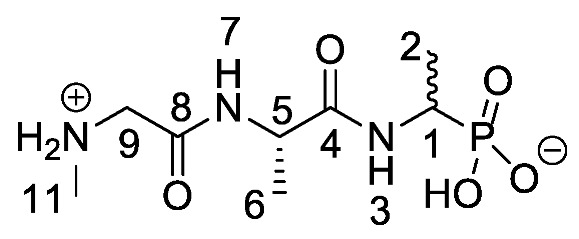


*(1-((S)-2-(2-(Methylamino)acetamido)propanamido)ethyl) phosphonic acid or Sar-l-Ala-d/l-Fos* (**21b**). The *tert*-butoxycarbonyl and diethyl ester protecting groups of (1-((*S*)-2-((*S*)-2-aminopropanamido)propanamido)ethyl)phosphonic acid (**18b**) (3.3 mmol, 1.40 g) were removed. The pale green crude solid was recrystallised from hot water/ethanol to give **21b** as a pale green solid composed of 2 diastereoisomers, Sar-l-Ala-l-Fos and Sar-l-Ala-D-Fos (0.45 g, 1.7 mmol, 51%); m.p. 241–245 °C (decomp.); *ῡ*_max_/cm^−1^ 3289 (NH^+^), 3500–2900 (br OH), 1632 (br C=O), 1556 (NH bend), 1174 (P=O), 1059 (P-O-C), 919 (P-OH); ^1^H NMR (300 MHz, D_2_O) δ_H_ 1.14–1.57 (6H, m, C**H_3_**-2, C**H_3_**-6), 2.74 (3H, s, C**H_3_**-11), 3.84–4.07 (3H, m, C**H_2_**-9, C**H**-1), 4.32–4.58 (1H, m, C**H**-5); ^13^C NMR (75 MHz, D_2_O) δ_C_ 15.4 (**C**H_3_-2), 16.8 (**C**H_3_-6), 32.9 (**C**H_3_-11), 43.9 (d, ^1^*J_P-C_* = 148.5 Hz, **C**H-1), 49.4 (**C**H_2_-9), 50.0 (**C**H-5), 166.0 (**C**=O-8), 173.7 (**C**=O-4); ^31^P-^1^H_decoup_ NMR (121 MHz, CDCl_3_) δ_P_ 19.2; HRMS (NSI) calcd for (C_8_H_19_N_3_O_5_P)^+^, MH^+^: 268.1057, found 268.1016; LCMS purity >95% (C-18 reversed phase, MeOH–H_2_O). (The LCMS chromatogram and conditions may be found within the Appendix A).


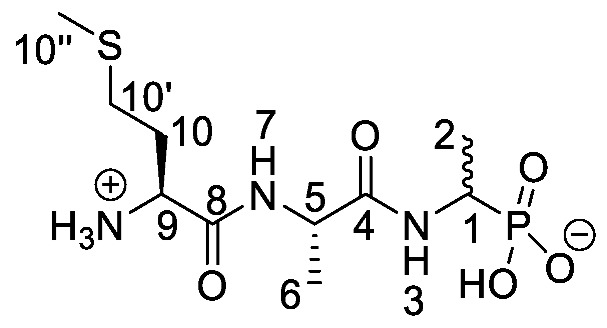


*(1-((S)-2-((S)-2-Amino-4-(methylthio)butanamido) propanamido)ethyl)phosphonic acid or l-Met-l-Ala-d/l-Fos* (**21c**). The *tert*-butoxycarbonyl and diethyl ester protecting groups of *tert*-butyl ((2*S*)-1-(((2*S*)-1-((1-(diethoxyphosphoryl)ethyl)amino)-1-oxopropan-2-yl)amino-4-(methylthio)-1-oxobutan-2-yl)carbamate (**18c**) (0.9 mmol, 0.43 g) were removed. The green crude solid was recrystallised from hot water/ethanol to give **21c** as a pale green solid composed of 2 diastereoisomers, l-Met-l-Ala-l-Fos and l-Met-l-Ala-D-Fos (0.13 g, 0.41 mmol, 46%); m.p. 214–217 °C (decomp.); *ῡ*_max_/cm^−1^ 3263 (NH^+^), 2834 (broad OH), 1641 (br C=O), 1552 (NH bend), 1150 (P=O), 1041 (P-O-C), 919 (P-OH); ^1^H NMR (300 MHz, D_2_O) δ_H_ 1.29–1.33 (3H, m, C**H_3_**-2), 1.42 (3H, d, ^3^*J*_H-H_ = 6.0 Hz, C**H_3_**-6), 2.15 (3H, s, C**H_3_**-10″), 2.20 (2H, m, C**H_2_**-10), 2.62 (2H, m, C**H_2_**-10′), 4.05 (1H, m, C**H**-1), 4.14 (1H, m, C**H**-9), 4.39–4.41 (1H, m, C**H**-5); ^13^C NMR (75 MHz, D_2_O) δ_C_ 16.9 (**C**H_3_-10″), 18.4 (**C**H_3_-2), 19.5 (**C**H_3_-6), 19.6 (**C**H_3_-6), 32.8 (**C**H_2_-10′), 33.0 (**C**H_2_-10′), 30.7 (**C**H_2_-10), 31.0 (**C**H_2_-10), 44.4 (**C**H-1), 52.9 (**C**H-5), 53.0 (**C**H-5), 55.0 (**C**H-9), 176.1 (**C**=O-4, **C**=O-8); ^31^P-^1^H_decoup_ NMR (121 MHz, CDCl_3_) δ_P_ 20.7; HRMS (NSI) calcd for (C_10_H_23_N_3_O_5_PS)^+^, MH^+^: 328.1091, found 328.1094; LCMS purity >95% (C-18 reversed phase, MeOH-H_2_O). (The LCMS chromatogram and conditions may be found within the Appendix A).


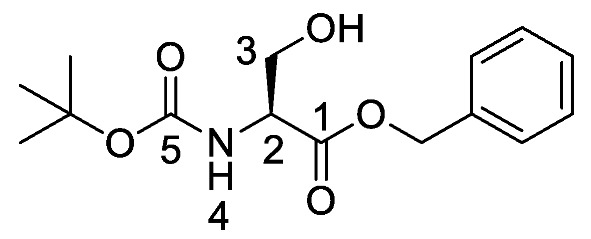


*(S)-Benzyl 2-((tert-butoxycarbonyl)amino)-3-hydroxy propanoate or Boc-l-Ser-OBzl*
**(11)**. Boc-l-Serine (**10**) (20 mmol, 4.10 g) and 1,8-diazabicyclo [5.4.0]undec-7-ene (DBU) (30 mmol, 4.5 mL) were dissolved in a round-bottom flask containing dry benzene (80 mL), followed by the addition of benzyl bromide (30 mmol, 3.60 mL). Caution: Benzene is a known carcinogen. The solution was stirred overnight at room temperature under nitrogen and later the solvent was removed under reduced pressure to afford an off-white residue. Ethyl acetate (100 mL) was added, the flask contents were sonicated and then washed with 1M HCl (50 mL) and brine (2 × 50 mL). The organic layer was dried over MgSO_4_, filtered, concentrated *in vacuo* and purified by column chromatography (petrol/ethyl acetate (1:1)) to give **11** as a white solid (5.24 g, 17.7 mmol, 89%); m.p. 61–63 °C (lit. m.p. 59–60 °C) [30]; *ῡ*_max_/cm^−1^ 3416 (NH, OH), 1756 (C=O), 1666 (C=O), 1522 (NH bend), 1200 (C-O), 1154 (C-O); ^1^H NMR (300 MHz, CDCl_3_) δ_H_ 1.36 (9H, s, C(C**H_3_**)**_3_**), 2.17 (1H, br, O**H**), 3.82 (1H, dd, ^2^*J*_H-H_ = 11.1 Hz, ^3^*J*_H-H_ = 3.6 Hz, C**H_a/b_**-3), 3.90 (1H, dd, ^2^*J*_H-H_ = 11.1 Hz, ^3^*J*_H-H_ = 3.9 Hz, C**H_a/b_**-3), 4.33 (1H, m, C**H**-2), 5.11 (1H, d, ^2^*J*_H-H_ = 12.3 Hz, OC**H_a/b_**Ar), 5.16 (1H, d, ^2^*J*_H-H_ = 12.3 Hz, OC**H_a/b_**Ar), 5.40 (1H, br, N**H**-4), 7.27 (5H, m, 5 × C**H_Ar_**); ^13^C NMR (75 MHz, CDCl_3_) δ_C_ 27.1 (C(**C**H_3_)_3_), 54.7 (**C**H-2), 62.3 (**C**H_2_-3), 66.2 (O**C**H_2_Ar), 79.1 (**C**(CH_3_)_3_), 127.0 (2 × **C**H_Ar_), 127.3 (**C**H_Ar_), 127.4 (2 × **C**H_Ar_), 134.1 (**C**H_Ar_ quat.), 153.0 (**C**=O-5), 170.7 (**C**=O-1); *m/z* (ESI) calcd for (C_15_H_21_NNaO_5_)^+^, MNa^+^: 318.3, found 318.2. (The ^1^H-NMR spectrum may be found within the Appendix A).


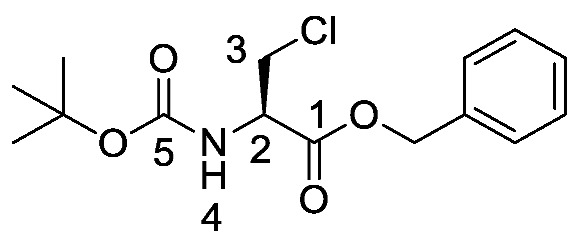


*(R)-Benzyl 2-((tert-butoxycarbonyl)amino)-3-chloropropanoate or Boc-β-Cl-l-Ala-OBzl* (**12**). (*S*)-Benzyl 2-((*tert*-butoxycarbonyl)amino)-3-hydroxypropanoate (**11**) (15 mmol, 4.43 g) was dissolved in dry DCM (40 mL), followed by the addition of trichloroacetonitrile (30 mmol, 3 ml). The solution was stirred at room temperature for 2 h. To this solution, triphenylphosphine (30 mmol, 7.87 g) in dry DCM (50 mL) was added slowly. The resulting solution was stirred overnight at room temperature under nitrogen; brine (100 mL) was added to quench the reaction. The organic layer was washed with brine (3 × 100 mL), dried over MgSO_4_, filtered and concentrated in vacuo to afford an orange residue. The residue was purified by column chromatography (petrol/ethyl acetate (7:3)) to give **12** as a white solid (3.53 g, 11.2 mmol, 75%); m.p. 55–58 °C; *ῡ*_max_/cm^−1^ 3364 (NH), 1725 (C=O), 1680 (C=O), 1519 (NH bend), 1208 (C-O), 1158 (C-O); ^1^H NMR (300 MHz, CDCl_3_) δ_H_ 1.38 (9H, s, C(C**H_3_**)**_3_**), 3.78 (1H, dd, ^2^*J*_H-H_ = 11.2 Hz, ^3^*J*_H-H_ = 3.2 Hz, C**H_a/b_**-3), 3.92 (1H, dd, ^2^*J*_H-H_ = 11.3 Hz, ^3^*J*_H-H_ = 3.0 Hz, C**H_a/b_**-3), 4.67 (1H, m, C**H**-2), 5.13 (1H, d, ^2^*J*_H-H_ = 12.2 Hz, OC**H_a/b_**Ar), 5.18 (1H, d, ^2^*J*_H-H_ = 12.2 Hz, OC**H_a/b_**Ar), 5.37 (1H, d, ^3^*J*_H-H_ = 7.5 Hz, N**H**-4), 7.29 (5H, m, 5 × C**H_Ar_**); ^13^C NMR (75 MHz, CDCl_3_) δ_C_ 28.3 (C(**C**H_3_)_3_), 45.5 (**C**H_2_-3), 54.5 (**C**H-2), 67.8 (O**C**H_2_Ar), 80.5 (**C**(CH_3_)_3_), 128.4 (**C**H_Ar_), 128.6 (**C**H_Ar_), 128.7 (**C**H_Ar_), 134.9 (**C**H_Ar_ quat.), 155.0 (**C**=O-5), 169.0 (**C**=O-1); *m/z* (ESI) calcd for (C_15_H_20_ClNNaO_4_)^+^, MNa^+^: 336.1 (^35^Cl), 338.1 (^37^Cl), found 336.2 (^35^Cl), 338.2 (^37^Cl); CHN (Found: C, 57.71; H, 6.46; N, 4.38. C_15_H_20_ClNO_4_ requires C, 57.42; H, 6.42; N, 4.46%). (The mass spectrum may be found within the Appendix A).


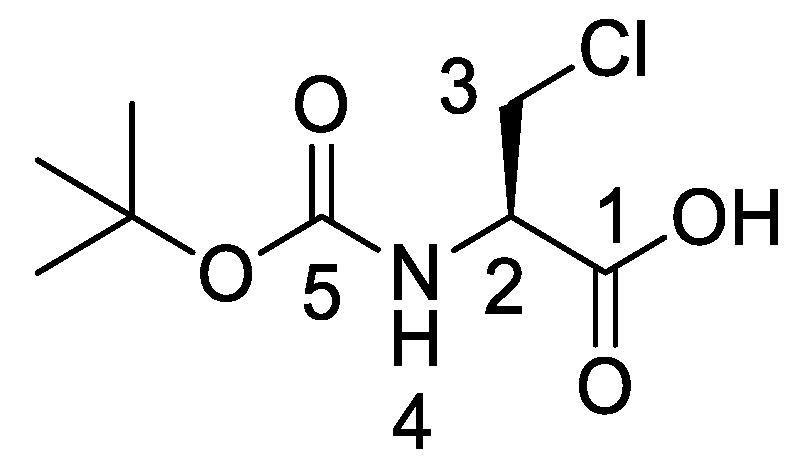


*(R)-2-((Tert-butoxycarbonyl)amino)-3-chloropropanoic acid or Boc-β-Cl-l-Ala-OH* (**13**). Deprotection of benzyl ester was followed, using (*R*)-benzyl 2-((*tert*-butoxycarbonyl)amino)-3-chloropropanoate (**12**) (7.0 mmol, 2.20 g) to afford **13** as an off-white solid (1.52 g, 6.78 mmol, 97%); m.p. 125–128 °C (lit. m.p. 127–129 °C) [15]; *ῡ*_max_/cm^−1^ 3434 (NH), 2973 (br OH), 1752 (C=O), 1735 (C=O), 1519 (NH bend), 1159 (C-O), 1148 (C-O); ^1^H NMR (300 MHz, CDCl_3_) δ_H_ 1.40 (9H, s, C(C**H_3_**)**_3_**), 3.80 (1H, dd, ^2^*J*_H-H_ = 12.0 Hz, ^3^*J*_H-H_ = 3.0 Hz, C**H_a/b_**-3), 3.95 (1H, dd, ^2^*J*_H-H_ = 12.0 Hz, ^3^*J*_H-H_ = 3.0 Hz, C**H_a/b_**-3), 4.70 (1H, m, C**H**-2), 5.42 (1H, d, ^3^*J*_H-H_ = 7.2 Hz, N**H**-4), 9.03 (1H, br, O**H**); ^13^C NMR (75 MHz, CDCl_3_) δ_C_ 27.1 (C(**C**H_3_)_3_), 44.0 (**C**H_2_-3), 53.1 (**C**H-2), 79.8 (**C**(CH_3_)_3_), 154.2 (**C**=O-5), 172.1 (**C**=O-1); *m/z* (ESI) calcd for (C_8_H_14_ClNNaO_4_)^+^, MNa^+^: 246.1 (^35^Cl), 248.1 (^37^Cl), found 246.1 (^35^Cl), 248.1 (^37^Cl).


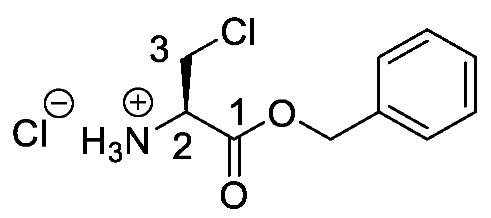


*(R)-1-(Benzyloxy)-3-chloro-1oxopropan-2-aminium chloride or β-Cl-l-Ala-OBzl hydrochloride* (**14**). Deprotection of *tert*-butoxycarbonyl was followed, using (*R*)-benzyl 2-((*tert*-butoxycarbonyl)amino)-3-chloropropanoate (**12**) (15 mmol, 4.71 g). The white crude solid was filtered and washed by diethyl ether to give **14** as a white solid (3.47 g, 13.4 mmol, 93%); m.p. 145 °C (sub); *ῡ*_max_/cm^−1^ 2841 (NH^+^), 1750 (C=O), 1231 (C-O); ^1^H NMR (300 MHz, D_2_O) δ_H_ 4.06 (1H, dd, ^2^*J*_H-H_ = 15.0 Hz, ^3^*J*_H-H_ = 6.0 Hz, C**H_a/b_**-3), 4.20 (1H, dd, ^2^*J*_H-H_ = 15.0 Hz, ^3^*J*_H-H_ = 6.0 Hz, C**H_a/b_**-3), 4.70 (1H, t, ^3^*J*_H-H_ = 6.0 Hz, C**H**-2), 5.29 (1H, d, ^2^*J*_H-H_ = 12.0 Hz, OC**H_a/b_**Ar), 5.37 (1H, d, ^2^*J*_H-H_ = 12.0 Hz, OC**H_a/b_**Ar), 7.42-7.47 (5H, m, 5 × C**H_Ar_**); ^13^C NMR (75 MHz, D_2_O) δ_C_ 41.8 (**C**H_2_-3), 54.0 (**C**H-2), 69.1 (O**C**H_2_Ar), 128.6–129.1 (**C**H_Ar_), 134.5 (**C**H_Ar_ quart.), 167.0 (**C**=O-1); *m/z* (ESI) calcd for (C_10_H_13_ClNO_2_), M^+^: 214.1 (^35^Cl), 216.1 (^37^Cl), found 214.1 (^35^Cl), 216.1 (^37^Cl); CHN (Found: C, 47.16; H, 5.43; N, 5.43. C_10_H_13_Cl_2_NO_2_∙0.2H_2_O requires C, 47.34; H, 5.32; N, 5.52%).


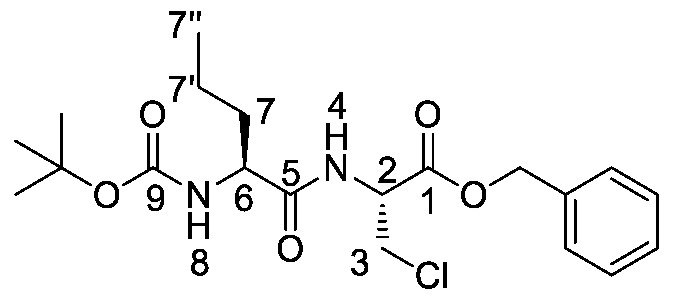


*(R)-Benzyl 2-((S)-2-((tert-butoxycarbonyl)amino)pentanamido)-3-chloropropanoate or Boc-l-Nva-β-chloro-l-Ala-OBzl* (**22a**). General peptide coupling method was followed, using Boc-l-Nva-OH (**15a**) (6.0 mmol, 1.31 g) in dry THF and (*R*)-1-(benzyloxy)-3-chloro-1oxopropan-2-aminium chloride (**14**) (5.4 mmol, 1.36 g) in dry DCM. The yellow crude liquid was purified by column chromatography (40-60 petrol/ethyl acetate (7:3)) to give **22a** as a white solid (1.74 g, 4.2 mmol, 78%); m.p. 95–98 °C; *ῡ*_max_/cm^−1^ 3327 (NH), 1743 (C=O), 1688 (C=O), 1653 (C=O), 1518 (NH bend), 1206 (C-O), 1169 (C-O); ^1^H NMR (300 MHz, CDCl_3_) δ_H_ 0.92 (3H, t, ^3^*J*_H-H_ = 9.0 Hz, C**H_3_**-7″), 1.32-1.43 (2H, m, C**H_2_**-7′), 1.45 (9H, s, C(C**H_3_**)_3_), 1.52–1.65 (1H, m, C**H_a/b_**-7), 1.75-1.82 (1H, m, C**H_a/b_**-7), 3.89 (1H, dd, ^2^*J*_H-H_ = 12.0 Hz, ^3^*J*_H-H_ = 3.0 Hz, C**H_a/b_**-3), 3.99 (1H, dd, ^2^*J*_H-H_ = 12.0 Hz, ^3^*J*_H-H_ = 3.0 Hz, C**H_a/b_**-3), 4.11–4.15 (1H, m, C**H**-6), 4.96–5.00 (2H, m, C**H**-2, N**H**-8), 5.20 (1H, d, ^2^*J*_H-H_ = 12.0 Hz, OC**H_a/b_**Ar), 5.25 (1H, d, ^2^*J*_H-H_ = 12.0 Hz, O**CH_a/b_**Ar), 6.97 (1H, d, ^3^*J*_H-H_ = 6.0 Hz, N**H**-4), 7.33–7.37 (5H, m, 5 x C**H_Ar_**); ^13^C NMR (75 MHz, CDCl_3_) δ_C_ 12.7 (**C**H_3_-7″), 17.8 (**C**H_2_-7′), 27.3 (C(**C**H_3_)_3_), 33.4 (**C**H_2_-7), 43.8 (**C**H_2_-3), 52.2 (**C**H-2), 53.4 (**C**H-6), 67.0 (O**C**H_2_Ar), 79.3 (**C**(CH_3_)_3_), 127.4 (**C**H_Ar_), 127.6 (**C**H_Ar_), 127.7 (**C**H_Ar_), 133.8 (**C**H_Ar_ quat.), 154.5 (**C**=O-9), 167.5 (**C**=O-1), 171.2 (**C**=O-5); HRMS (NSI) calcd for (C_20_H_30_ClN_2_O_5_)^+^, MH^+^: 413.1838 (^35^Cl), 415.1809 (^37^Cl), found 413.1837 (^35^Cl), 415.1807 (^37^Cl); CHN (Found: C, 58.49; H, 7.22; N, 6.81. C_20_H_29_ClN_2_O_5_ requires C, 58.18; H, 7.08; N, 6.78%).


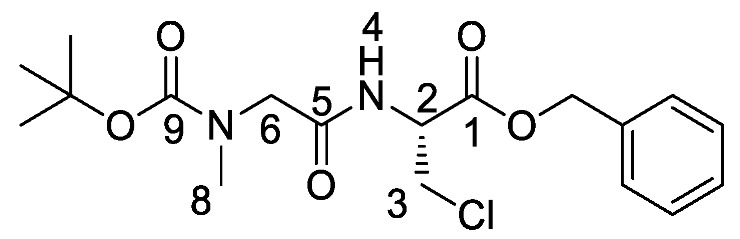


*(R)-Benzyl 2-(2-((tert-butoxycarbonyl)(methyl)amino) acetamido)-3-chloropropanoate or Boc-Sar-β-chloro-l-Ala-OBzl* (**22b**). General peptide coupling method was followed, using Boc-Sar-OH (**15b**) (13.0 mmol, 2.46 g) in dry THF and (*R*)-1-(benzyloxy)-3-chloro-1oxopropan-2-aminium chloride (**14**) (13.3 mmol, 3.34 g) in dry DCM. The yellow crude liquid was purified by column chromatography (40–60 petrol/ethyl acetate (7:3)) to give **22b** as a light yellow syrup (3.63 g, 9.4 mmol, 73%); *ῡ*_max_/cm^−1^ 3302 (NH), 1747 (C=O), 1686 (br C=O), 1522 (NH bend), 1175 (C-O), 1148 (C-O); ^1^H NMR (300 MHz, CDCl_3_) δ_H_ 1.40 (9H, s, C(C**H_3_**)_3_), 2.87 (3H, s, NC**H_3_**-8), 3.80 (1H, d, ^2^*J*_H-H_ = 15.0 Hz, C**H_a/b_**-6), 3.82 (1H, d, ^2^*J*_H-H_ = 15.0 Hz, C**H_a/b_**-6), 3.83 (1H, dd, ^2^*J*_H-H_ = 12.0 Hz, ^3^*J*_H-H_ = 3.0 Hz, C**H_a/b_**-3), 3.94 (1H, dd, ^2^*J*_H-H_ = 12.0 Hz, ^3^*J*_H-H_ = 3.0 Hz, C**H_a/b_**-3), 4.91–4.96 (1H, m, C**H**-2), 5.13 (1H, d, ^2^*J*_H-H_ = 12.0 Hz, OC**H_a/b_**Ar), 5.18 (1H, d, ^2^*J*_H-H_ = 12.0 Hz, O**CH_a/b_**Ar), 6.97 (1H, d, ^3^*J*_H-H_ = 6.0 Hz, N**H**-4), 7.26–7.30 (5H, m, 5 x C**H_Ar_**); ^13^C NMR (75 MHz, CDCl_3_) δ_C_ 28.2 (C(**C**H_3_)_3_), 35.6 (N**C**H_3_-8), 44.9 (**C**H_2_-3), 53.0 (**C**H_2_-6), 53.0 (**C**H-2), 68.0 (O**C**H_2_Ar), 81.0 (**C**(CH_3_)_3_), 128.4 (**C**H_Ar_), 128.6 (**C**H_Ar_), 128.7 (**C**H_Ar_), 134.8 (**C**H_Ar_ quat.), 154.5 (**C**=O-9), 168.4 (**C**=O-1), 169.4 (**C**=O-5); HRMS (NSI) calcd for (C_18_H_26_ClN_2_O_5_)^+^, MH^+^: 385.1525 (^35^Cl), 387.1496 (^37^Cl), found 385.1527 (^35^Cl), 387.1498 (^37^Cl). LCMS purity >95% (C-18 reversed phase, MeOH–H_2_O).


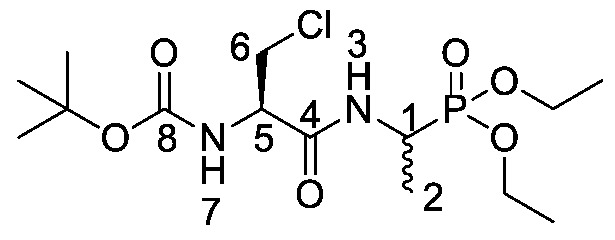


*Tert-butyl ((2R)-3-chloro-1-((1-(diethoxyphosphoryl)ethyl) amino)-1-oxopropan-2-yl)carbamate or Boc-β-chloro-l-Ala-d/l-Fos diethyl ester* (**19e**). General peptide coupling method was followed, using (*R*)-2-((tert-butoxycarbonyl)amino)-3-chloropropanoic acid (**13**) (6.0 mmol, 1.34 g) in dry THF and diethyl 1-aminoethylphosphonate (**9**) (6.0 mmol, 1.09 g) in dry THF. The light yellow crude liquid was purified by column chromatography, using 100% petrol and increasing to 100% ethyl acetate, to afford **19e** as colorless syrup composed of 2 diastereoisomers, Boc-β-Cl-l-Ala-l-Fos diethyl ester and Boc-β-Cl-l-Ala-D-Fos diethyl ester (2.03 g, 5.2 mmol, 88%); ῡ_max_/cm^−1^ 3261 (NH), 1713 (C=O), 1670 (C=O), 1517 (NH bend), 1225 (P=O), 1164 (C-O), 1020 (P-O-C), 970 (P-O-C); ^1^H NMR (300 MHz, CDCl_3_) δ_H_ 1.15–1.46 (9H, m, C**H_3_**-2, 2 × OCH_2_C**H_3_**), 1.47 (9H, s, C(C**H_3_**)_3_), 3.74 (1H, dd, ^2^*J*_H-H_ = 12.0 Hz, ^3^*J*_H-H_ = 6.0 Hz, C**H_a/b_**-6), 4.00 (1H, dd, ^2^*J*_H-H_ = 12.0 Hz, ^3^*J*_H-H_ = 6.0 Hz, C**H_a/b_**-6), 4.06–4.22 (4H, m, 2 × OC**H_2_**CH_3_), 4.40–4.56 (2H, m, C**H**-1, C**H**-5), 5.46 (0.5H, d, ^3^*J*_H-H_ = 6.0 Hz, N**H**-3 or N**H**-7), 5.48 (0.5H, d, ^3^*J*_H-H_ = 9.0 Hz, N**H**-3 or N**H**-7), 7.01 (0.5H, m, N**H**-3 or N**H**-7), 7.09 (0.5H, m, N**H**-3 or N**H**-7); ^13^C NMR (75 MHz, CDCl_3_) δ_C_ 15.6 (**C**H_3_-2), 15.7 (**C**H_3_-2), 16.3 (d, ^3^*J*_C-P_ = 1.5 Hz, OCH_2_**C**H_3_), 16.4 (d, ^3^*J*_C-P_ = 2.3 Hz, OCH_2_**C**H_3_), 16.4 (d, ^3^*J*_C-P_ = 1.5 Hz, OCH_2_**C**H_3_), 16.4 (d, ^3^*J*_C-P_ = 2.3 Hz, OCH_2_**C**H_3_), 28.2 (C(**C**H_3_)_3_), 41.2 (d, ^1^*J*_C-P_ = 157.5 Hz, **C**H-1), 41.3 (d, ^1^*J*_C-P_ = 157.5 Hz, **C**H-1), 55.2 (**C**H_2_-6), 55.3 (**C**H-5), 62.6 (d, ^2^*J*_C-P_ = 6.8 Hz, O**C**H_2_CH_3_), 62.6 (d, ^2^*J*_C-P_ = 6.8 Hz, O**C**H_2_CH_3_), 63.0 (d, ^2^*J*_C-P_ = 6.8 Hz, O**C**H_2_CH_3_), 63.0 (d, ^2^*J*_C-P_ = 6.8 Hz, O**C**H_2_CH_3_), 80.8 (**C**(CH_3_)_3_), 155.0 (**C**=O-8), 168.3 (**C**=O-4); ^31^P-^1^H_decoup_ NMR (121 MHz, CDCl_3_) δ_P_ 24.7, 24.8; HRMS (NSI) calcd for (C_14_H_28_ClN_2_O_6_P), MH^+^: 409.1266 (^35^Cl), 411.1237 (^37^Cl), found 409.1258 (^35^Cl), 411.1231 (^37^Cl). LCMS purity >95% (C-18 reversed phase, MeOH-H_2_O).


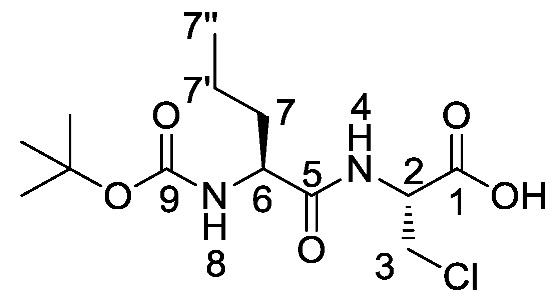


*(R)-2-((S)-2-((tert-Butoxycarbonyl)amino)pentanamido)-3-chloropropanoic acid or Boc-l-Nva-β-chloro-l-Ala-OH* (**23a**). Deprotection of benzyl ester was followed, using (*R*)-benzyl 2-((*S*)-2-((*tert*-butoxycarbonyl)amino)pentanamido)-3-chloropropanoate (**22a**) (5.8 mmol, 2.41 g) to afford **23a** as a light yellow solid (1.81 g, 5.61 mmol, 96%); m.p. 60–63 °C; *ῡ*_max_/cm^−1^ 3312 (br OH), 2963 (NH), 1655 (br C=O), 1509 (NH bend), 1161 (C-O); ^1^H NMR (300 MHz, DMSO) δ_H_ 0.85 (3H, t, ^3^*J*_H-H_ = 9.0 Hz, C**H_3_**-7″), 1.24–1.34 (2H, m, C**H_2_**-7′), 1.38 (9H. s, C(C**H_3_**)_3_), 1.42–1.52 (1H, m, C**H_a/b_**-7), 1.54–1.59 (1H, m, C**H_a/b_**-7), 3.34 (1H, br, O**H**), 3.84 (1H, dd, ^2^*J*_H-H_ = 12.0 Hz, ^3^*J*_H-H_ = 6.0 Hz, C**H_a/b_**-3), 3.91 (1H, dd, ^2^*J*_H-H_ = 12.0 Hz, ^3^*J*_H-H_ = 6.0 Hz, C**H_a/b_**-3), 3.95–4.02 (1H, m, C**H**-6), 4.62–4.67 (1H, m, C**H**-2), 6.92 (1H, d, ^3^*J*_H-H_ = 9.0 Hz, N**H**-8), 8.07 (1H, d, ^3^*J*_H-H_ = 9.0 Hz, N**H**-4); ^13^C NMR (75 MHz, CDCl_3_) δ_C_ 14.1 (**C**H_3_-7″), 19.1 (**C**H_2_-7′), 28.6 (C(**C**H_3_)_3_), 34.4 (**C**H_2_-7), 45.1 (**C**H_2_-3), 53.6 (**C**H-2), 54.5 (**C**H-6), 78.5 (**C**(CH_3_)_3_), 155.8 (**C**=O-9), 170.6 (**C**=O-5), 173.0 (**C**=O-1); *m/z* (ESI) calcd for (C_13_H_23_ClN_2_NaO_5_)^+^, MNa^+^: 345.1 (^35^Cl), 347.1 (^37^Cl), found 345.2 (^35^Cl), 347.2 (^37^Cl); CHN (Found: C, 48.67; H, 7.51; N, 8.42. C_13_H_23_ClN_2_O_5_ requires C, 48.37; H, 7.18; N, 8.68%).


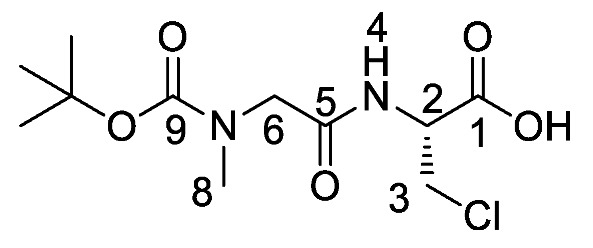


*(R)-2-(2-((tert-butoxycarbonyl)(methyl)amino)acetamido)-3-chloropropanoic acid or Boc-Sar-β-chloro-l-Ala-OH* (**23b**). Deprotection of benzyl ester was followed, using (*R*)-benzyl 2-(2-((*tert*-butoxycarbonyl)(methyl)amino)acetamido)-3-chloropropanoate (**22b**) (6.2 mmol, 2.40 g) to afford **23b** as an off-white solid (1.81 g, 6.1 mmol, 99%); m.p. 89–91 °C; *ῡ*_max_/cm^−1^ 3342 (NH), 2982 (br OH), 1734 (C=O), 1672 (C=O), 1644 (C=O), 1524 (NH bend), 1152 (C-O); ^1^H NMR (300 MHz, CDCl_3_) δ_H_ 1.47 (9H. s, C(C**H_3_**)_3_), 2.99 (3H, s, NC**H_3_**-8), 3.81–4.17 (4H, m, C**H_2_**-6, C**H_2_**-3), 5.01 (1H, m, C**H**-2), 7.07 (1H, br, N**H**-4), 7.45 (1H, br, O**H**); ^13^C NMR (75 MHz, CDCl_3_) δ_C_ 28.3 (C(**C**H_3_)_3_), 36.1 (N**C**H_3_-8), 44.5 (**C**H_2_-3), 53.0 (**C**H_2_-6 and **C**H-2), 81.8 (**C**(CH_3_)_3_), 156.8 (**C**=O-9), 169.5 (**C**=O-1 and **C**=O-5); *m/z* (ESI) calcd for (C_11_H_19_ClN_2_NaO_5_)^+^, MNa^+^: 317.1 (^35^Cl), 319.1 (^37^Cl), found 317.1 (^35^Cl), 319.1 (^37^Cl); CHN (Found: C, 43.98; H, 6.69; N, 9.53. C_11_H_19_ClN_2_O_5_·0.3H_2_O requires C, 44.02; H, 6.58; N, 9.33%).


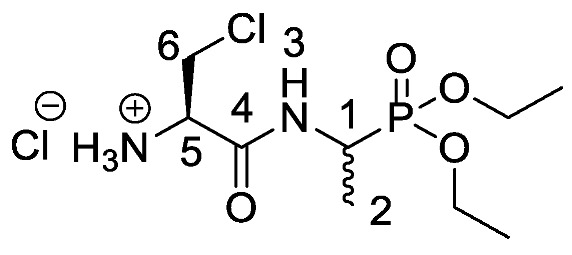


*(2R)-3-Chloro-1-((1-(diethoxyphosphoryl)ethyl)amino)-1-oxopropan-2-aminium chloride or β-Cl-l-Ala-d/l-Fos diethyl ester hydrochloride* (**20e**). Deprotection of tert-butoxycarbonyl was followed, using ((2R)-3-chloro-1-((1-(diethoxyphosphoryl)ethyl)amino)-1-oxopropan-2-yl)carbamate (**19e**) (6.7 mmol, 2.59 g). The off-white hygroscopic crude solid was washed with petrol to afford **20e** as a pale green solid composed of 2 diastereoisomers, β-Cl-l-Ala-l-Fos diethyl ester hydrochloride and β-Cl-l-Ala-D-Fos diethyl ester hydrochloride (1.51 g, 4.7 mmol, 70%); m.p. 129–133 °C (decomp.); ῡ_max_/cm^−1^ 3204 (NH^+^), 1687 (C=O), 1562 (NH bend), 1204 (P=O), 1010 (P-O-C), 961 (P-O-C); ^1^H NMR (300 MHz, D_2_O) δ_H_ 1.28 (3H, t, ^3^*J*_H-H_ = 6.0 Hz, OCH_2_C**H_3_**), 1.29 (3H, t, ^3^*J*_H-H_ = 6.0 Hz, OCH_2_C**H_3_**), 1.37 (3H, dd, ^3^*J*_P-H_ = 18.0 Hz, ^3^*J*_H-H_ = 6.0 Hz, C**H_3_**-2), 3.92–4.04 (2H, m, C**H_2_**-6), 4.07-4.21 (4H, m, 2 × OC**H_2_**CH_3_), 4.38–4.48 (2H, m, C**H**-1, C**H**-5); ^13^C NMR (75 MHz, D_2_O) δ_C_ 13.7 (**C**H_3_-2), 14.0 (**C**H_3_-2), 15.7 (OCH_2_**C**H_3_), 15.7 (OCH_2_**C**H_3_), 41.7 (d, ^1^*J*_P-C_ = 158.3 Hz, **C**H-1), 42.0 (d, ^1^*J*_P-C_ = 157.5 Hz, **C**H-1), 42.4 (**C**H_2_-6), 53.7 (**C**H-5), 53.8 (**C**H-5), 64.3 (d, ^2^*J*_P-C_ = 6.8 Hz, O**C**H_2_CH_3_), 64.5 (d, ^2^*J*_P-C_ = 6.8 Hz, O**C**H_2_CH_3_), 165.7 (**C**=O-4), 165.8 (**C**=O-4); ^31^P-^1^H_decoup_ NMR (121 MHz, CDCl_3_) δ_P_ 26.1, 26.2; HRMS (NSI) calcd for (C_9_H_21_ClN_2_O_4_P)^+^, M^+^: 287.0922 (^35^Cl), 289.0892 (^37^Cl), found 287.0922 (^35^Cl), 289.0890 (^37^Cl). LCMS purity >95% (C-18 reversed phase, MeOH–H_2_O).


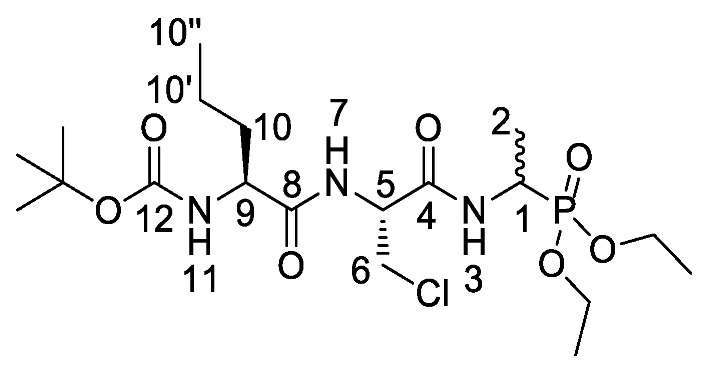


*Tert-butyl ((2S)-1-(((2R)-3-chloro-1-((1-(diethoxyphosphoryl)-ethyl)amino)-1-oxopropan-2-yl)amino)-1-oxopentan-2-yl)carbamate or Boc-l-Nva-β-chloro-l-Ala-d/l-Fos diethyl ester* (**24a**). General peptide coupling method was followed, using (*R*)-2-((S)-2-((tert-butoxycarbonyl)amino)pentanamido)-3-chloropropanoic acid (**23a**) (1.8 mmol, 0.58 g) in dry THF and diethyl 1-aminoethylphosphonate (**9**) (1.8 mmol, 0.33 g) in dry THF. The light yellow crude liquid was purified by column chromatography, using ethyl acetate/methanol (96:4), to afford **24a** as a white solid composed of 2 diastereoisomers, Boc-l-Nva-β-Cl-l-Ala-l-Fos diethyl ester and Boc-l-Nva-β-Cl-l-Ala-D-Fos diethyl ester (0.45 g, 0.93 mmol, 52%); m.p. 196 °C (decomp); ῡ_max_/cm^−1^ 3272 (NH), 1709 (C=O), 1680 (C=O), 1644 (C=O), 1530 (NH bend), 1229 (P=O), 1165 (C-O), 1019 (P-O-C), 972 (P-O-C); ^1^H NMR (300 MHz, CDCl_3_) δ_H_ 0.86 (1.5H, t, ^3^*J*_H-H_ = 9.0 Hz, C**H_3_**-10″), 0.88 (1.5H, t, ^3^*J*_H-H_ = 9.0 Hz, C**H_3_**-10″), 1.22–1.34 (11H, m, 2 × OCH_2_C**H_3_**, C**H_3_**-2, C**H_2_**-10′), 1.38 (9H, s, C(C**H_3_**)_3_), 1.53-1.59 (1H, m, C**H_a/b_**-10), 1.70–1.77 (1H, m, C**H_a/b_**-10), 3.69 (1H, dd, ^2^*J*_H-H_ = 12.0 Hz, ^3^*J*_H-H_ = 6.0 Hz, C**H_a/b_**-6), 3.78-3.81 (1H, m, C**H**-9), 3.91 (1H, dd, ^2^*J*_H-H_ = 12.0 Hz, ^3^*J*_H-H_ = 6.0 Hz, C**H_a/b_**-6), 3.97-4.13 (4H, m, 2 x OC**H_2_**CH_3_), 4.35-4.46 (1H, m, C**H**-1), 4.73-4.79 (1H, m, C**H**-5), 4.97–5.03 (1H, m, N**H**-11), 7.01 (0.5H, d, ^3^*J*_H-H_ = 9.0 Hz, N**H**-7), 7.09 (0.5H, d, ^3^*J*_H-H_ = 9.0 Hz, N**H**-7), 7.25 (0.5H, d, ^3^*J*_H-H_ = 9.0 Hz, N**H**-3), 7.33 (0.5H, d, ^3^*J*_H-H_ = 9.0 Hz, N**H**-3); ^13^C NMR (75 MHz, CDCl_3_) δ_C_ 12.7 (**C**H_3_-10″), 14.5 (**C**H_3_-2), 15.3 (OCH_2_**C**H_3_), 15.4 (OCH_2_**C**H_3_), 15.5 (OCH_2_**C**H_3_), 15.6 (OCH_2_**C**H_3_), 17.9 (**C**H_2_-10′), 18.0 (**C**H_2_-10′), 27.0 (C(**C**H_3_)_3_), 27.3 (C(**C**H_3_)_3_), 33.2 (**C**H_2_-10), 40.4 (d, ^1^*J*_C-P_ = 157.5 Hz, **C**H-1), 43.4 (**C**H_2_-6), 52.6 (**C**H-5), 52.8 (**C**H-5), 61.4 (d, ^2^*J*_C-P_ = 6.8 Hz, O**C**H_2_CH_3_), 61.6 (d, ^2^*J*_C-P_ = 6.8 Hz, O**C**H_2_CH_3_), 61.7 (d, ^2^*J*_C-P_ = 6.8 Hz, O**C**H_2_CH_3_), 61.9 (d, ^2^*J*_C-P_ = 6.8 Hz, O**C**H_2_CH_3_), 70.5 (**C**H-9), 79.4 (**C**(CH_3_)_3_), 154.7 (**C**=O-12), 166.8 (**C**=O-4), 171.4 (**C**=O-8); ^31^P-^1^H_decoup_ NMR (121 MHz, CDCl_3_) δ_P_ 24.9, 25.0; HRMS (NSI) calcd for (C_19_H_38_ClN_3_O_7_P)^+^, MH^+^: 486.2130 (^35^Cl), 488.2102 (^37^Cl), found 486.2124 (^35^Cl), 488.2098 (^37^Cl); CHN (Found: C, 46.61; H, 7.76; N, 8.31. C_19_H_37_ClN_3_O_7_P requires C, 46.96; H, 7.67; N, 8.65%).


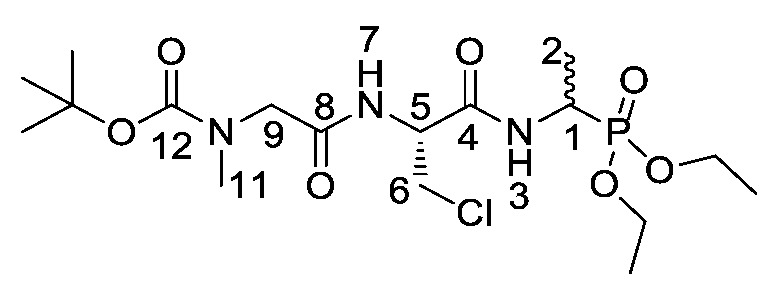


*Tert-butyl (2-(((2R)-3-chloro-1-((1-(diethoxyphosphoryl)ethyl) amino)-1-oxopropan-2-yl)amino)-2-oxoethyl)(methyl)carbamate or Boc-Sar-β-chloro-l-Ala-d/l-Fos diethyl ester* (**24b**). General peptide coupling method was followed, using (*R*)-2-(2-((tert-butoxycarbonyl)(methyl)amino)acetamido)-3-chloropropanoic acid (**23b**) (5.5 mmol, 1.61 g) in dry THF and diethyl 1-aminoethylphosphonate (**9**) (6.0 mmol, 1.09 g) in dry THF. The light yellow crude liquid was purified by column chromatography, using DCM/methanol (95:5), to afford **24b** as a light yellow syrup composed of 2 diastereoisomers, Boc-Sar-β-Cl-l-Ala-l-Fos diethyl ester and Boc-Sar-β-Cl-l-Ala-D-Fos diethyl ester (1.93 g, 4.21 mmol, 76%); ῡ_max_/cm^−1^ 3218 (NH), 1690 (C=O), 1665 (br C=O), 1518 (NH bend), 1224 (P=O), 1148 (C-O), 1018 (P-O-C), 967 (P-O-C); ^1^H NMR (300 MHz, CDCl_3_) δ_H_ 1.11–1.35 (9H, m, 2 × OCH_2_C**H_3_**, C**H_3_**-2), 1.41 (9H, s, C(C**H_3_**)_3_), 2.90 (3H, s, C**H_3_**-11), 3.70–3.88 (4H, m, C**H_2_**-6, C**H_2_**-9), 4.02–4.13 (4H, m, 2 × OC**H_2_**CH_3_), 4.36–4.47 (1H, m, C**H**-1), 4.78–4.82 (1H, m, C**H**-5), 6.94 (1H, m, N**H**-7), 7.36 (1H, m, N**H**-3); ^13^C NMR (75 MHz, CDCl_3_) δ_C_ 15.2 (**C**H_3_-2), 15.6 (**C**H_3_-2), 16.3 (OCH_2_**C**H_3_), 16.4 (OCH_2_**C**H_3_), 28.3 (C(**C**H_3_)_3_), 35.9 (**C**H_3_-11), 41.2 (d, ^1^*J*_C-P_ = 156.8 Hz, **C**H-1), 44.7 (**C**H_2_-6), 53.1 (**C**H_2_-9), 53.4 (**C**H-5), 62.7 (d, ^2^*J*_C-P_ = 6.0 Hz, O**C**H_2_CH_3_), 63.0 (d, ^2^*J*_C-P_ = 7.5 Hz, O**C**H_2_CH_3_), 81.0 (**C**(CH_3_)_3_), 152.3 (**C**=O-12), 167.7 (**C**=O-4 or **C**=O-8), 169.4 (**C**=O-4 or **C**=O-8); ^31^P-^1^H_decoup_ NMR (121 MHz, CDCl_3_) δ_P_ 24.7, 24.8; HRMS (NSI) calcd for (C_16_H_34_ClN_3_O_7_P)^+^, MH^+^: 480.1637 (^35^Cl), 482.1608 (^37^Cl), found 480.1642 (^35^Cl), 482.1612 (^37^Cl). LCMS purity >92% (C-18 reversed phase, MeOH-H_2_O).


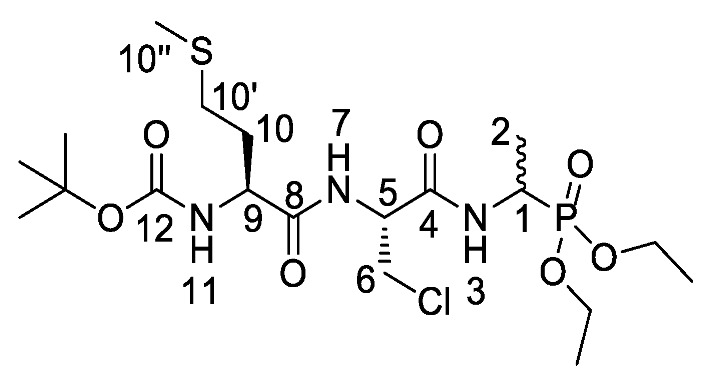


*Tert-butyl ((2S)-1-(((2R)-3-chloro-1-((1-(diethoxyphosphoryl) ethyl)amino)-1-oxopropan-2-yl)amino)-4-(methylthio)-1-oxobutan-2-yl) carbamate or Boc-l-Met-β-Cl-l-Ala-d/l-Fos diethyl ester* (**24c**). General peptide coupling method was followed, using Boc-l-Met-OH (**15c**) (3.4 mmol, 0.85 g) in dry THF and (2R)-3-chloro-1-((1-(diethoxyphosphoryl)ethyl)amino)-1-oxopropan-2-aminium chloride (**20e**) (3.4 mmol, 1.10 g) in dry DCM. The yellow crude liquid was purified by column chromatography (DCM/MeOH (95:5)) and recrystallized from diethyl ether/petrol to give **24c** as a white solid composed of 2 diastereoisomers, Boc-l-Met-β-Cl-l-Ala-l-Fos diethyl ester and Boc-l-Met-β-Cl-l-Ala-D-Fos diethyl ester (0.88 g, 1.7 mmol, 50%); m.p. 96–99 °C; ῡ_max_/cm^−1^ 3278 (NH), 1709 (C=O), 1687 (C=O), 1639 (C=O), 1523 (NH bend), 1228 (P=O), 1165 (C-O), 1018 (P-O-C), 970 (P-O-C); ^1^H NMR (300 MHz, CDCl_3_) δ_H_ 1.17-1.36 (9H, m, C**H_3_**-2, 2 × OCH_2_C**H_3_**), 1.38 (9H, s, C(C**H_3_**)_3_), 1.87–2.03 (2H, m, C**H_2_**-10), 2.04 (3H, s, C**H_3_**-10″), 2.48–2.54 (2H, m, C**H_2_**-10′), 3.71 (1H, dd, ^2^*J*_H-H_ = 12.0 Hz, ^3^*J*_H-H_ = 6.0 Hz, C**H_a/b_**-6), 3.88 (1H, dd, ^2^*J*_H-H_ = 12.0 Hz, ^3^*J*_H-H_ = 6.0 Hz, C**H_a/b_**-6), 3.99–4.13 (4H, m, 2 × OC**H_2_**CH_3_), 4.20 (1H, m, C**H**-9), 4.37–4.47 (1H, m, C**H**-1), 4.78–4.84 (1H, m, C**H**-5), 5.39 (0.5H, d, ^3^*J*_H-H_ = 6.0 Hz, N**H**-11), 5.41 (0.5H, d, ^3^*J*_H-H_ = 6.0 Hz, N**H**-11), 7.15 (0.5H, d, ^3^*J*_H-H_ = 6.0 Hz, N**H**-7), 7.24 (0.5H, d, ^3^*J*_H-H_ = 6.0 Hz, N**H**-7), 7.52 (1H, m, N**H**-3); ^13^C NMR (75 MHz, CDCl_3_) δ_C_ 14.3 (**C**H_3_-2), 14.4 (**C**H_3_-2), 14.5 (**C**H_2_-10″), 15.3 (OCH_2_**C**H_3_), 15.4 (OCH_2_**C**H_3_), 15.5 (OCH_2_**C**H_3_), 15.6 (OCH_2_**C**H_3_), 27.3 (C(**C**H_3_)_3_), 29.2 (**C**H_2_-10′), 29.3 (**C**H_2_-10′), 30.2 (**C**H_2_-10), 30.4 (**C**H_2_-10), 40.3 (d, ^1^*J*_P-C_ = 159.0 Hz, **C**H-1), 43.5 (**C**H_2_-6), 43.7 (**C**H_2_-6), 52.7 (**C**H-5), 53.1 (**C**H-9), 61.6 (d, ^2^*J*_P-C_ = 6.8 Hz, O**C**H_2_CH_3_), 61.7 (d, ^2^*J*_P-C_ = 6.0 Hz, O**C**H_2_CH_3_), 62.0 (d, ^2^*J*_P-C_ = 6.8 Hz, O**C**H_2_CH_3_), 62.1 (d, ^2^*J*_P-C_ = 7.5 Hz, O**C**H_2_CH_3_), 79.6 (**C**(CH_3_)_3_), 154.8 (**C**=O-12), 166.7 (**C**=O-4), 166.8 (**C**=O-4), 170.7 (**C**=O-8), 170.8 (**C**=O-8); ^31^P-^1^H_decoup_ NMR (121 MHz, CDCl_3_) δ_P_ 24.5, 24.8; HRMS (NSI) calcd for (C1_9_H_3H_ClN_3_O_7_PS)^+,^ MH^+^: 518.1851 (^35^Cl), 520.1821 (^37^Cl), found 518.1842 (^35^Cl), 520.1814 (^37^Cl); CHN (Found: C, 44.08; H, 7.47; N, 8.18. C_19_H_37_ClN_3_O_7_PS requires C, 44.06; H, 7.20; N, 8.11%).


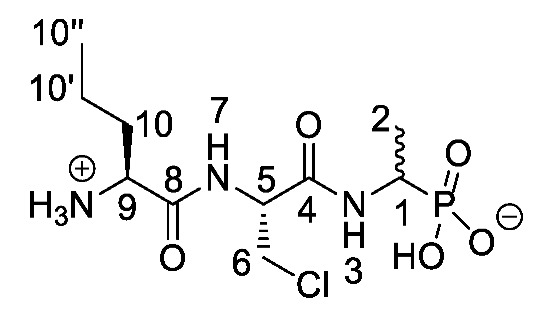


*(1-((R)-2-((S)-2-Ammoniopentanamido)-3-chloropropanamido) ethyl)phosphonic acid or l-Nva-β-chloro-l-Ala-d/l-Fos* (**25a**). The *tert*-butoxycarbonyl and diethyl ester protecting groups of *tert*-butyl ((2*S*)-1-(((2*R*)-3-chloro-1-((1-(diethoxyphosphoryl)-ethyl)amino)-1-oxopropan-2-yl)amino)-1-oxopentan-2-yl)carbamate (**24a**) (2.0 mmol, 0.99 g) were removed. The pale green crude solid was washed with diethyl ether to give **25a** as a pale green solid composed of 2 diastereoisomers, l-Nva-β-Cl-l-Ala-l-Fos and l-Nva-β-Cl-l-Ala-D-Fos (0.64 g, 1.94 mmol, 97%); m.p. 175 °C (sub); *ῡ*_max_/cm^−1^ 3294 (NH^+^), 3000 (br OH), 1668 (C=O), 1645 (C=O), 1538 (NH bend), 1132 (P=O), 1039 (P-O-C), 921 (P-OH); ^1^H NMR (300 MHz, D_2_O) δ_H_ 1.01 (3H, t, ^3^*J*_H-H_ = 9.0 Hz, C**H_3_**-10″), 1.30–1.37 (3H, br m, C**H_3_**-2), 1.44–1.54 (2H, br m C**H_2_**-10′), 1.90–1.98 (2H, br m, C**H_2_**-10), 3.91–4.15 (4H, br m, C**H_2_**-6, C**H**-9, C**H**-1), 4.79 (1H, br m, C**H**-5); ^13^C NMR (75 MHz, D_2_O) δ_C_ 12.9 (**C**H_3_-10″), 15.7 (**C**H_3_-2), 17.6 (**C**H_2_-10′), 33.0 (**C**H_2_-10), 43.3 (**C**H_2_-6), 53.1 (**C**H-1 and **C**H-9), 55.0 (**C**H-5), 170.4 (**C**=O-4 and **C**=O-8); ^31^P-^1^H_decoup_ NMR (121 MHz, CDCl_3_) δ_P_ 18.5; HRMS (NSI) calcd for (C_10_H_20_ClN_3_O_5_P)^-,^ MH^-^: 328.0835 (^35^Cl), 330.0805 (^37^Cl), found 328.0833 (^35^Cl), 330.0800 (^37^Cl). LCMS purity >95% (C-18 reversed phase, MeOH-H_2_O). (The LCMS chromatogram and conditions may be found within the Appendix A).


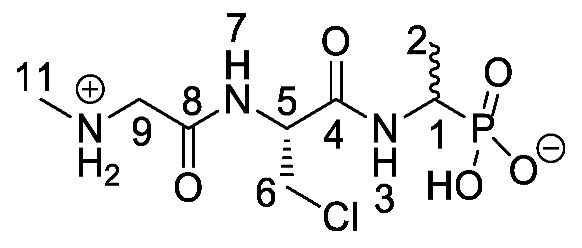


*1-((R)-3-Chloro-2-(2-(methylammonio)acetamido) propanamido)ethyl)phosphonic acid or Sar-β-chloro-l-Ala-d/l-Fos* (**25b**). The *tert*-butoxycarbonyl and diethyl ester protecting groups of *tert*-butyl (2-(((2*R*)-3-chloro-1-((1-(diethoxyphosphoryl)ethyl)amino)-1-oxopropan-2-yl)amino)-2-oxoethyl)(methyl)carbamate (**24b**) (3.8 mmol, 1.74 g) were removed. The pale green crude solid was recrystallised from hot water/ethanol to give **25b** as an off-white solid composed of 2 diastereoisomers, Sar-β-Cl-l-Ala-l-Fos and Sar-β-Cl-l-Ala-D-Fos (0.49 g, 1.61 mmol, 42%); m.p. 185-188 °C (decomp.); *ῡ*_max_/cm^−1^ 3287 (NH^+^), 3000 (br OH), 1657 (C=O), 1634 (C=O), 1552 (NH bend), 1172 (P=O), 1054 (P-O-C), 919 (P-OH); ^1^H NMR (300 MHz, CD_3_OD) δ_H_ 1.24 (3H, dd, ^3^*J*_H-P_ = 15.0 Hz, ^3^*J*_H-H_ = 6.0 Hz, C**H_3_**-2), 2.74 (3H, s, NC**H_3_**-11), 3.81–3.87 (2H, m, C**H_2_**-6), 3.93–3.94 (2H, m, C**H_2_**-9), 3.97–4.10 (1H, m, C**H**-1), 4.79 (1H, m, C**H**-5); ^13^C NMR (75 MHz, CD_3_OD) δ_C_ 15.4 (**C**H_3_-2), 32.9 (N**C**H_3_-11), 43.6 (**C**H_2_-6), 44.1 (d, ^1^*J*_C-P_ = 148.5 Hz, **C**H-1), 49.5 (**C**H_2_-9), 54.6 (**C**H-5), 166.4 (**C**=O-4 or **C**=O-9), 166.9 (**C**=O-4 or **C**=O-9); ^31^P-^1^H_decoup_ NMR (121 MHz, CDCl_3_) δ_P_ 18.8; HRMS (NSI) calcd for (C_8_H_18_ClN_3_O_5_P)^+,^ MH^+^: 302.0667 (^35^Cl), 304.0638 (^37^Cl), found 302.0670 (^35^Cl), 304.0640 (^37^Cl). LCMS purity >95% (C-18 reversed phase, MeOH-H_2_O). (The LCMS chromatogram and conditions may be found within the Appendix A).


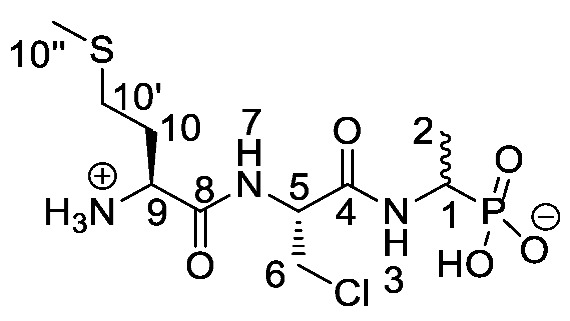


*(1-((R)-2-((S)-2-Ammonio-4-(methylthio)butanamido)-3-chloro propanamido)ethyl)phosphonic acid or l-Met-β-Cl-l-Ala-d/l-Fos* (**25c**). The tert-butoxycarbonyl and diethyl ester protecting groups of tert-butyl ((2S)-1-(((2R)-3-chloro-1-((1-(diethoxyphosphoryl) ethyl)amino)-1-oxopropan-2-yl)amino)-4-(methylthio)-1-oxobutan-2-yl)carbamate (**24c**) (1.4 mmol, 0.71 g) were removed. The green crude solid was recrystallised from hot water/ethanol to give **25c** as a pale green solid composed of 2 diastereoisomers, l-Met-β-Cl-l-Ala-l-Fos and l-Met-β-Cl-l-Ala-D-Fos (0.17 g, 0.48 mmol, 35%); m.p. 175–179 °C (decomp.); ῡ_max_/cm^−1^ 3264 (NH^+^), 2829 (broad OH), 1666(C=O), 1641 (C=O), 1546 (NH bend), 1149 (P=O), 1041 (P-O-C), 921 (P-OH); ^1^H NMR (300 MHz, D_2_O) δ_H_ 1.31 (3H, dd, ^3^*J*_H-P_ = 15.0 Hz, ^3^*J*_H-H_ = 6.0 Hz, C**H_3_**-2), 2.13 (3H, s, C**H_3_**-10″), 2.18–2.29 (2H, m, C**H_2_**-10), 2.63–2.69 (2H, m, C**H_2_**-10′), 3.89 (1H, dd, ^2^*J*_H-H_ = 12.0 Hz, ^3^*J*_H-H_ = 6.0 Hz, C**H_a/b_**-6), 3.97 (1H, dd, ^2^*J*_H-H_ = 12.0 Hz, ^3^*J*_H-H_ = 6.0 Hz, C**H_a/b_**-6), 4.01–4.13 (1H, m, C**H**-1), 4.22 (1H, br m, C**H**-9), 4.75–4.79 (1H, m, C**H**-5); ^13^C NMR (75 MHz, D_2_O) δ_C_ 16.9 (**C**H_3_-10″), 17.0 (**C**H_3_-10″), 18.4 (**C**H_3_-2), 31.1 (**C**H_2_-10′), 32.9 (**C**H_2_-10), 46.2 (**C**H_2_-6), 47.0 (d, ^1^*J*_C-P_ = 147.0 Hz, **C**H-1), 52.2 (**C**H-9), 52.3 (**C**H-9), 57.8 (**C**H-5), 58.0 (**C**H-5), 171.7 (**C**=O-4), 171.8 (**C**=O-4), 172.3 (**C**=O-8); ^31^P-^1^H_decoup_ NMR (121 MHz, CDCl_3_) δ_P_ 18.7; HRMS (NSI) calcd for (C_10_H_21_ClN_3_O_5_PS), MNa^+^: 384.0520 (^35^Cl), 386.0489 (^37^Cl), found 384.0523 (^35^Cl), 386.0491 (^37^Cl). LCMS purity >95% (C-18 reversed phase, MeOH-H_2_O). (The LCMS chromatogram and conditions may be found within the Appendix A).

### 3.2. Microbiological Procedures

#### 3.2.1. Media Constituents

l-Arginine, l-aspartic acid, l-cysteine, glycine, l-histidine, l-isoleucine, l-lysine, l-methionine, l-phenylalanine, l-proline, l-serine, l-threonine, l-tryptophan, l-valine, magnesium sulphate, haemin and d-(+)-glucose were purchased from Sigma Chemical Co. (Poole, UK). l-Tyrosine, uracil, guanine, cytosine, adenine, ammonium sulphate, potassium dihydrogen phosphate, saponin, and dipotassium hydrogen phosphate were acquired from BDH Merck Ltd. (Poole, England). Yeast extract was supplied by bioMérieux (Craponne, France) and bacteriological agar from Oxoid (Basingstoke, UK). Nicotinamide adenine dinucleotide (NAD) was obtained from Merck (Darmstadt, Germany) and heparinized horse blood from TCS Biosciences (Buckingham, UK).

#### 3.2.2. Microbiology Strains

Microbial reference strains were obtained from National Collection of Type Cultures (NCTC) (Colindale, UK) and the American Type Culture Collection (ATCC) (Manassas, US). These included twelve Gram-negative bacteria: *A. baumannii*, *B. cepacia*, *E. cloacae*, *E. coli* (*n* = 2), *K. pneumoniae*, *P. rettgeri*, *P. aeruginosa*, *S. typhimurium*, *S. enteritidis*, *S. marcescens,* and *Y. enterocolitica*; as well as seven Gram positive bacteria: *E. faecalis*, *E. faecium*, *L. monocytogenes*, *S. epidermidis*, *S. aureus*, methicillin resistant *S. aureus* (MRSA), and *S. pyogenes*. These 19 bacterial strains were maintained on Columbia agar.

#### 3.2.3. Preparation of Antagonist-free (AF) Medium

The methodology utilized, which was reported by Atherton et al. [11], was adopted with slight modification. In this method, 1.5% (15 g/L) bacteriological agar and 0.5% (5 g/L) glucose were added to 880 mL deionized water. The resulting mixture was autoclaved at 121 °C for 15 min, followed by addition of 2 %v/v (20 mL/L) saponin-lysed horse blood, 25 mg/L hemin, 25 mg/L NAD and 100 mL of 10 x strength antagonist-free broth prepared as previously described [11]. The saponin-lysed blood was prepared by incubating 100 mL sterile hose blood at 37 ± 0.5 °C for 15 min, adding 5 mL 10% saponin and re-incubating for 15 min or longer to ensure completely blood lysis. The saponin-lysed blood was stored at 4 °C until used. The pH of medium was 7.0.

#### 3.2.4. Preparation of Media Containing Phosphonotripeptide Derivatives

Each derivative was dissolved in sterile deionized water (SDW) and incorporated with AF-agar at a concentration range of 0.032–8 mg/L. A series of dilutions was performed to achieve the required concentration and then poured into the sterile Petri dishes. Plates containing AF-agar were also prepared without inhibitor. Solidified agar plates were placed in a warm cabinet (37 ± 0.5 °C) for 5 min to dry the surface of the agar and then stored at 4 °C.

#### 3.2.5. Multiple Inoculation of Agar

Each microbial strain was isolated from the respective Columbia agar after 18 h of incubation and suspended in SDW to a density equivalent to 0.5 McFarland units using a densitometer. One hundred microliters of each suspension was transferred into the corresponded wells of a multipoint inoculation device. One microliter of the bacterial suspension, equivalent to 1.5 × 10^5^ organisms was applied as a spot by this instrument on the plates containing different phosphonotripeptide derivatives (as well as inhibit-free growth control plates). Nineteen different bacterial strains were applied on a plate and incubated for 22 h at 37 ± 0.5 °C. Un-inoculated AF-agar plates from the same batch of agar were incubated as a sterility check and a sterility check was also performed on the SDW used to make bacterial suspensions by culture of aliquots on Columbia agar.

#### 3.2.6. Determination of Minimum Inhibitory Concentration (MIC)

After incubation, each plate was inspected for growth or inhibition of the inoculated spots. The minimum inhibitory concentration (MIC) was recorded as the lowest concentration of inhibitor to completely inhibit the growth of the test strain. Tests were performed in duplicate on separate occasions and in the vast majority of cases generated identical results. In a small minority of cases, results differed but were always within one double-dilution and this was always resolved by a third test. MIC values were only interpreted for any particular strain if growth on the inhibitor-free control plate was completely uninhibited and if sterility of the culture medium and diluent were validated.

## 4. Conclusions

Here we have disclosed the synthesis of six phosphonopeptide based inhibitors, all of which contain the antimicrobial agent fosfalin at their *C*-terminus. Three of the inhibitors also contain a second antimicrobial agent, β-chloroalanine, at the center of their tripeptide sequence. The identity of the *N*-terminal amino acid was varied in order to elicit selectivity via exploiting differences in permeability and hydrolysis rates within bacterial cells.

We have shown that the β-chloro containing compounds elicit lower MIC values than their alanine analogues against a wide range of clinically relevant bacteria. This is consistent with release of free β-chloroalanine which then is able to act synergistically with the fosfalin unit.

Finally, we show that the MIC profiles of inhibitors **25a** and **25c** have potential for two differing real-world clinical applications. These are the detection of lung pathogens in cystic fibrosis patients, and the detection of *Salmonella* within stool samples for diagnosis of food poisoning, However, further work is required with large numbers of bacterial strains and relevant clinical samples to conclusively demonstrate the utility of these compounds.

## 5. Patents

Antimicrobial Compounds, 2018, WO2015140481.

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
