# Peer review of "Synthesis and Antimicrobial Activity of Phosphonopeptide Derivatives Incorporating Single and Dual Inhibitors"

_molecules, 2020, doi:10.3390/molecules25071557_

Round 1

Reviewer 1 Report

This manuscript by Keng Tiong Ng and coworkers describes the synthesis and microbiological evaluation of a series of tripeptide derivatives containing D,L-fosfalin at C-terminal, either Ala or βCl-Ala at the central position, and 3 different N-terminal amino acids.

It is difficult to assess the real interest/potential of these tripeptides as antimicrobial agents because no comparison with model, simpler compounds 1,2 and 4 has been done.

Synthesis: Two alternative strategies have been used in the preparation of tripeptide derivatives. The couplings were performed either using a dipeptide free carboxylic acid (18a,b and 24a,b) or a free amino dipeptide (18c and 24c). Could you comment about the reason for doing in that manner? Is this related to the N-terminal residue in the tripeptide? A table including the yield of all compounds, and the ratio of diastereoisomers could be highly informative.

The numbering in scheme 3 is rather confusing. Compound 20d led to 18c, 20e led to 24c. The use of letters does not help in this scheme. Different compounds must have different numbers.

Activity: MIC values are given in mg/L, which do not allow a good comparison among compounds with different molecular weight, as it is the case. These values should be indicated in terms of concentration (mM, µM, etc).

The potential future applications of the described compounds (las paragraph in section 2.3) are rather speculative, not based on experimental results. In this respect, also the abstract does not give clear idea of the content of the manuscript, but erroneously incites to think that there were applications in diagnostic microbiology. Similarly occurs in the last paragraph in conclusions. Therefore, these parts need for a deep rewriting.

It would be desirable to attempt the separation of at least one diastereoisomeric mixture to know the influence of configuration (if any).

It is surprising that the [α]20D has been measured for some racemic compounds (value 0) but not for enantiopure amino acid and dipetide derivatives.

A caution with the use of benzene should be indicated.

Figure S5, S7 and S8, apart from the main peaks, other minor components (most probably diastereoisomers) can be appreciated. For a peptide specialist this could be anticipated, since the epimerization of the C-terminal residue most probably will take place during the coupling reaction of dipeptide free carboxylates. This should be commented within the text.

Figure S7. The second compound is 25b (only 25 is indicated)

Experimental: you should choose between extended or short nomenclature, there is no reason for using both. If you prefer the the abbreviated nomenclature, use Boc instead tBoc

A list of abbreviations used should be convenient.

Authors stated no conflict of interest. However, as the research is funded by a corporation, the author should issue a public statement that the research is free of bias.

Author Response

This manuscript by Keng Tiong Ng and coworkers describes the synthesis and microbiological evaluation of a series of tripeptide derivatives containing D,L-fosfalin at C-terminal, either Ala or βCl-Ala at the central position, and 3 different N-terminal amino acids.

It is difficult to assess the real interest/potential of these tripeptides as antimicrobial agents because no comparison with model, simpler compounds 1,2 and 4 has been done.

Comments from authors: The simpler compounds 1, 2 and 4 and their antimicrobial properties have been assessed against different strains/types of bacteria. These results are available and cited in the manuscript (ref. 11-13 & ref. 15-16). The key point within this current paper is not the overall potency against all bacteria, it is the ability of our compounds to display selectivity in their action. It is this selectivity that produces their utility as diagnostic tools within clinical microbiology. We would like to reiterate here that the compounds disclosed in this manuscript are not intended to be used as therapeutic antimicrobial agents.

Synthesis: Two alternative strategies have been used in the preparation of tripeptide derivatives. The couplings were performed either using a dipeptide free carboxylic acid (18a,b and 24a,b) or a free amino dipeptide (18c and 24c). Could you comment about the reason for doing in that manner? Is this related to the N-terminal residue in the tripeptide? A table including the yield of all compounds, and the ratio of diastereoisomers could be highly informative.

Comments from authors: An explanation for the differing strategies has been included on lines 130-134.

In addition, a summary of yields of all compounds and the ratio of diastereoisomers has now been incorporated in supplementary Table S1, and is referred to in the main manuscript in lines 354-355.

The numbering in scheme 3 is rather confusing. Compound 20d led to 18c, 20e led to 24c. The use of letters does not help in this scheme. Different compounds must have different numbers.

Comments from authors: Scheme 3 is a comprehensive summary of the synthesis of our phosphonotripeptide derivatives, with different letters representing different side chains attached at R1 and R2. We have tried to keep the letter designation consistent throughout the scheme, so that each letter donates a particular core backbone. The authors acknowledge that different compounds should normally be assigned different compound numbers. However, in this case as the phosphonotripeptide derivatives share a common backbone, with only different substituents at R1 and R2, the authors believe that assigning each compound a different number would result in unnecessary repetition of compound structures that will increase the complexity to the reader rather than decrease it. However, if upon due reflection the reviewers and editors would like the numbering scheme to be changed we will indeed try our best to accede to this request.

Activity: MIC values are given in mg/L, which do not allow a good comparison among compounds with different molecular weight, as it is the case. These values should be indicated in terms of concentration (mM, µM, etc).

Comments from authors: International methods for determination of the minimum inhibitory concentration are highly prescriptive and all such methods insist that the potency of antimicrobials should be assessed in terms of mg/L. Furthermore this must be done based on log2 dilutions of 1 mg/L (e.g. 0.065, 0.125, 0.25, 0.5,1,2, etc….). Hence we felt there was no alternative but to adhere to this standard practice. This also has the advantage of allowing comparisons with previously published data for related fosfalin derivatives – which all follow this same practice.

We accept of course that this limits our ability to compare the potency of the compounds reported herein but this is very much of secondary importance. Of primary importance is the inhibitory profile of the compounds – for example a large difference in MIC between pathogens (e.g. Salmonella) and commensals (e.g. E. coli).

Whilst it would obviously be possible to quote all values in mmol/L as well as mg/L, this would make the manuscript very messy as the range in mmol/L would be different for every compound. We feel this would lead to unnecessary confusion and would depart from the practice of all previous papers that have examined inhibitors of this type.

The potential future applications of the described compounds (last paragraph in section 2.3) are rather speculative, not based on experimental results. In this respect, also the abstract does not give clear idea of the content of the manuscript, but erroneously incites to think that there were applications in diagnostic microbiology. Similarly occurs in the last paragraph in conclusions. Therefore, these parts need for a deep rewriting.

Comments from authors: Our experimental results clearly show good and selective inhibition of compound 25a and 25c against certain commensal bacteria as stated in Table 1. The purpose of such compounds in the context of clinical diagnostic microbiology is to prevent the overgrowth of commensal bacteria that can camouflage the presence of bacterial pathogens of interest. In this case our compounds inhibit the growth of commensal organisms allowing specific pathogens to grow thus facilitating their detection. It appears counterintuitive at first glance but the goal of this work is to NOT inhibit key pathogens.

Despite this, we agree with the reviewer that the results are preliminary with respect to the potential of these compounds for diagnostic use and that much more work is required in follow-up studies. We have therefore added the following text to the end of section 2.3 (lines 216-224):

In this paper we present preliminary microbiology data to identify novel compounds that may prove useful as novel selective agents in diagnostic culture media. A major limitation of our data is that only 1 or 2 examples of any particular species have been tested and susceptibility (or resistance) may not be uniformly demonstrated across all strains of a particular species. The utility of these compounds can only be proven in subsequent studies by testing large numbers of strains from each relevant species and large numbers of clinical samples to see if the isolation of target pathogens may be enhanced. However, the fact that such utility as selective agents has been demonstrated for other peptide mimetics based on fosfalin provides encouragement that such agents may ultimately prove useful [27, 28].

We have also added two additional references here that demonstrate the proven diagnostic utility of related compounds.

We do not feel that the abstract requires modification as it merely states that the compounds “show potential” for diagnostic application and we have now stated clearly the limitations of our work. We have however, ‘toned down’ our conclusions by adding a final sentence stating that “However, further work is required with large numbers of bacterial strains and relevant clinical samples to conclusively demonstrate the utility of these compounds.” Lines 348-350.

It would be desirable to attempt the separation of at least one diastereoisomeric mixture to know the influence of configuration (if any).

Comments from authors: New text added to lines 90-96. In our previous unpublished investigations we have seen that whenever the enanotimerically pure form of fosfalin has been utilized within a peptide sequence the activity is always twice that of the analogous compound containing the racemic version of fosfalin. This is in keeping with the notion that one isomer of fosfalin is active and the other is completely inactive. As the intended use in the present work is to produce economically viable diagnostic tools rather than potent antimicrobial agents, we elected to utilize the racemic version of this warhead within this study.

It is surprising that the [α]20D has been measured for some racemic compounds (value 0) but not for enantiopure amino acid and dipetide derivatives.

Comments from authors: Optical rotation was only performed on fosfalin to ensure that it was indeed fully racemic prior to peptide coupling. The authors did not perform optical rotation on other dipeptide derivatives for several reasons. Firstly, the amino acids used for coupling were either enantiomeric pure or synthesised from enantiomeric pure starting materials. Secondly the amount of compound required to conduct accurate [α]20D was usually more than we synthesised in a given batch. We would be reticent in quoting values that could be deemed unreliable. If it aids in the clarity and consistency of the manuscript we could remove the values that are already included, as these values are not typically deemed essential for publication for peptide derivatives.

A caution with the use of benzene should be indicated.

Comments from authors: “Caution: Benzene is a known carcinogen” has been added to line 117.

Figure S5, S7 and S8, apart from the main peaks, other minor components (most probably diastereoisomers) can be appreciated. For a peptide specialist this could be anticipated, since the epimerization of the C-terminal residue most probably will take place during the coupling reaction of dipeptide free carboxylates. This should be commented within the text.

Comments from authors: We agree with the reviewer about epimerisation. Minor components were not detected in 21a, 21b and 21c, instead they were visible in 25a, 25b and 25c, i.e. only in the compounds with a central beta-chloroalanine residue. We believe that the presence of a good leaving group in the form of the chlorine atom, which could leave as chloride via an elimination process at the β-position of the alanine would facilitate the epimerisation process, explain g the differences seen in the chromatograms. Comments have been added in the supplementary material within Figures S7 and S8.

Figure S7. The second compound is 25b (only 25 is indicated)

Comments from authors: Figure S7 has been rectified.

[α]20D

Experimental: you should choose between extended or short nomenclature, there is no reason for using both. If you prefer the abbreviated nomenclature, use Boc instead tBoc

Comments from authors: The tBoc abbreviation has been replaced with Boc throughout the manuscript and supplementary material.

A list of abbreviations used should be convenient.

Comments from authors: A list of abbreviations has been included as Table S1 in the supplementary material. A note directing the reader to this list has been added to the main manuscript on line 28 and is again mentioned in line 354.

Authors stated no conflict of interest. However, as the research is funded by a corporation, the author should issue a public statement that the research is free of bias.

Comments from authors: The following statement has been added to lines 364-366 “The funders had no role in the design of the study; in the collection, analyses, or interpretation of data; in the writing of the manuscript, or in the decision to publish the results.”

Reviewer 2 Report

Authors synthesized and characterised six novel small peptides and tested its susceptibility for various bacterial strains. Interesting results indicates potentional clinical application of some derivatives in the process of pathogen cultivation and identification.

Synthesis and structural characterization of compounds is precisely described. Identity and purity of compounds is thoroughly documented in supplementary information.

Microbiological procedures are described with less accuracy.

Information how positive and negative control were performed have to be added.

How exactly the MIC was determined? Were the plates evaluated in triplicate? The MIC value is the average value or the highest value observed?

Please add above mentioned information into pargraph 3.2.6

After completing this paragraph the article will be worthy for publication.

Author Response

Authors synthesized and characterised six novel small peptides and tested its susceptibility for various bacterial strains. Interesting results indicates potentional clinical application of some derivatives in the process of pathogen cultivation and identification.

Synthesis and structural characterization of compounds is precisely described. Identity and purity of compounds is thoroughly documented in supplementary information.

Microbiological procedures are described with less accuracy.

Information how positive and negative control were performed have to be added.

How exactly the MIC was determined? Were the plates evaluated in triplicate? The MIC value is the average value or the highest value observed?

Please add above mentioned information into pargraph 3.2.6

After completing this paragraph the article will be worthy for publication.

Comments from authors:

We agree with the reviewer that this section would benefit from further details.

In section 3.2.4 we now state that “Plates containing AF-agar were also prepared without inhibitor.” This is the positive (growth) control. Lines 314-315.

In section 3.2.5, we now state that “Un-inoculated AF-agar plates from the same batch of agar were incubated as a sterility check and a sterility check was also performed on the SDW used to make bacterial suspensions by culture of aliquots on Columbia agar”. Lines 324-326.

Finally section 3.2.6 has been expanded as follows:

“After incubation, each plate was inspected for growth or inhibition of the inoculated spots. The minimum inhibitory concentration (MIC) was recorded as the lowest concentration of inhibitor to completely inhibit the growth of the test strain. Tests were performed in duplicate on separate occasions and in the vast majority of cases generated identical results. In a small minority of cases, results differed but were always within one double-dilution and this was always resolved by a third test. MIC values were only interpreted for any particular strain if growth on the inhibitor-free control plate was completely uninhibited and if sterility of the culture medium and diluent were validated”. Lines 328-335.

 As these compounds are completely novel it was not possible to include a bacterial strain with a known MIC.

Round 2

Reviewer 1 Report

The manuscript is improved and convincing explanations have been given for most concerns.  Therefore, it can be published in Molecules after a few minor points:

L128-131: A number should be given for the reference. Compounds 20d and 20e shoul be in bold. A full stop has to be added after the parenthesis.

Supporting Information: Boc must neither be superscripted nor italized

Numbering of compounds: I leave this at the Editor choice.

 [α]20D : please, remove the 0 values.

The explanation about the epimerization because the chlorine atom seems not convincing, because if Cl atom acts as a good leaving group the resulting products will be the dehydroAla derivatives. This could easily be corroborated just by making HPLC-MS experiments.

Author Response

Reviewer’s comments (2nd round)

The manuscript is improved and convincing explanations have been given for most concerns.  Therefore, it can be published in Molecules after a few minor points:

L128-131: A number should be given for the reference. Compounds 20d and 20e shoul be in bold. A full stop has to be added after the parenthesis.

Author Comments: A new reference [21] was added in the reference section, corresponding to L128-131. Compounds 20d and 20e have been changed to bold. A full stop has been added after the parenthesis.

Supporting Information: Boc must neither be superscripted nor italized

Author Comments: Boc abbreviation has been changed as requested throughout the supplementary materials.

Numbering of compounds: I leave this at the Editor choice.

 [α]20D : please, remove the 0 values.

Author Comments: “[α]20D 0 values” have been removed from the supplementary information.

The explanation about the epimerization because the chlorine atom seems not convincing, because if Cl atom acts as a good leaving group the resulting products will be the dehydroAla derivatives. This could easily be corroborated just by making HPLC-MS experiments.

Author Comments: We thank this reviewer for this very interesting suggestion, and if it were feasible we would indeed like to corroborate the identity of substances the lie behind the minor peaks observed for each beta-chloro Ala containing tripeptide. Unfortunately, we are not in a position to conduct this additional experiment within the 5 days allotted to us for replying to these comments. There two key reasons for this:

  • The work outlined in this paper was held under an IP embargo for quite a few years. The samples are now over 5 years old, and thus significant degradation is likely to have occurred within them.
  • The key researcher that conducted the work at the time, Dr Keng Ng no longer works at our institution. He is currently a Postdoctoral Researcher at King’s College London and is not in a position to return to us to run the suggested experiment at the present time.

Thus, were we to attempt the suggested experiment we would need to ask another person to run these samples, and perhaps indeed completely resynthesise the tripeptides from scratch (see comment 1), which could take several months and many thousands of pounds of additional funding, and would perhaps lead to issues relating to additional authors on the paper. It is important to note that the main peaks make up 95% of each sample, and thus the purity of the samples meets expectations with regards to publication.

However, what we can say is that we did run an extensive NMR based stability study on the related compound betaChloroAla-Fosfalin. This work showed that over time, in an aqueous environment, slow degradation of the beta chloro unit within this substance did indeed occur and that the products obtained most likely were produced via the type of vinylic intermediate that the reviewer has suggested.

We have removed the comment that speculates as to the identity of these minor impurities.